**Data Availability Statement:** We have provide a link to the data, which as been published on a

# Evaluation of a large-scale flow manipulation to the upper San Francisco Estuary: Response of habitat conditions for an endangered native fish

Ted Sommer[1]*, Rosemary Hartman[1], Michal Koller[1], Michael Koohafkan[1], J. Louise Conrad[2], Michael MacWilliams[3], Aaron Bever[3], Christina Burdi[4], April Hennessy[4], Michael Beakes[5]

1 California Department of Water Resources, Sacramento, California, United States of America, 2 Delta Science Program, Sacramento, California, United States of America, 3 Anchor QEA, LLC, San Francisco, California, United States of America, 4 California Department of Fish and Wildlife, Stockton, California, United States of America, 5 U.S. Bureau of Reclamation, Sacramento, California, United States of America

* Ted.Sommer@water.ca.gov

## Abstract

While flow is known to be a major driver of estuarine ecosystems, targeted flow manipulations are rare because tidal systems are extremely variable in space and time, and because the necessary infrastructure is rarely available. In summer 2018 we used a unique water control structure in the San Francisco Estuary (SFE) to direct a managed flow pulse into Suisun Marsh, one of the largest contiguous tidal marshes on the west coast of the United States. The action was designed to increase habitat suitability for the endangered Delta Smelt *Hypomesus transpacificus*, a small osmerid fish endemic to the upper SFE. The approach was to operate the Suisun Marsh Salinity Control Gates (SMSCG) in conjunction with increased Sacramento River tributary inflow to direct an estimated $160 \times 10^6$ m$^3$ pulse of low salinity water into Suisun Marsh during August, a critical time period for juvenile Delta Smelt rearing. Three-dimensional modeling showed that directing additional low salinity water into Suisun Marsh ("Flow Action") substantially increased the area of low salinity habitat for Delta Smelt that persisted beyond the period of SMSCG operations. Field monitoring showed that turbidity and chlorophyll were at higher levels in Suisun Marsh, representing better habitat conditions, than the upstream Sacramento River region throughout the study period. The Flow Action had no substantial effects on zooplankton abundance, nor did Suisun Marsh show enhanced levels of these prey species in comparison to the Sacramento River. Fish monitoring data suggested that small numbers of Delta Smelt colonized Suisun Marsh from the Sacramento River during the 2018 Flow Action. Comparison of the salinity effects of the Flow Action to historical catch data for Suisun Marsh further supported our hypothesis that the Flow Action would have some benefit for this rare species. Our study provides insight into both the potential use of targeted flow manipulations to support endangered fishes such as Delta Smelt, and into the general response of estuarine habitat to flow management.

public repository, EDI: Hartman, R.K., T. Sommer, M. Koohafkan, C. Burdi, A. Bever, M. MacWilliams, and J. Galef. 2020. Interagency Ecological Program: Water quality, fish, and zooplankton monitoring and modeling to support the 2018 Suisun Marsh Salinity Control Gates Summer Action ver 2. Environmental Data Initiative. https://doi.org/10.6073/pasta/72d4abd5c679260d0130655d1179e47b (Accessed 2020-08-04).

**Funding:** The Lead Author's agency (California Department of Water Resources) provided support in the form of funding to a private consulting firm (Anchor QEA) to support salaries for two authors [MM, AB]. As the study lead for the project, Department of Water Resources therefore had a central role in the study design, data collection and analysis, decision to publish, and preparation of the manuscript. The specific roles of these authors are articulated in the 'author contributions' section."

**Competing interests:** Two of the authors (MM, AB) are employed by a private consulting firm. We confirm that this (commercial affiliation) does not alter our adherence to PLOS ONE policies on sharing data and materials.

# Introduction

Estuaries represent one of the single most challenging ecosystems for large scale experimental flow manipulations [1]. Such flow manipulations are moderately common in riverine locations, typically generated through changes in upstream dam operations [2, 3]. While there is substantial information about the response of estuaries to other management changes such as habitat restoration or development projects [4–6], it is rare that estuarine flow is the target of an experimental evaluation. A key reason for the paucity of estuarine flow experiments is that this ecosystem type has extreme variability, including tidal, seasonal, and annual variation in hydrology, as well as substantial habitat heterogeneity across their landscape. As a consequence, the estuarine response to experimental flow manipulations is complicated to evaluate [1]. Moreover, estuaries often lack the engineering infrastructure such as dams that allow for experimental flow manipulations on rivers or streams. For this reason, inflow effects on estuaries are often examined based on long-term monitoring of the responses of different ecological variables to a suite of hydrologic regimes or natural flow conditions [7], rather than targeted flow manipulations. A notable exception is an evaluation by Strydom and Whitfield [8] of the response of the Krommer Estuary (South Africa) to a regulated freshwater flow pulse. In addition, it is relatively common for bar-built estuaries to be artificially breached to reduce flood effects, improve water quality via introduction of oceanic flows, or to facilitate fish migration [9–11].

Perhaps nowhere else in the world have the effects of estuarine inflow been studied as much as in the San Francisco Estuary (SFE) [7, 12]. The SFE is one of the largest estuaries on the Pacific Coast, and includes downstream bays (San Francisco, San Pablo, Suisun) and a freshwater tidal delta formed by the confluence of the Sacramento and San Joaquin Rivers (Fig 1). The SFE and its tributaries have been heavily modified by many factors including urbanization, levee construction, water diversions, seasonal barriers, and levees to control water flow and levels in the delta, and dams to manage inflow from the tributaries. Numerous studies of the long-term effects of flow in the San Francisco Estuary (SFE) have provided insights into the response of a suite of physical, chemical, and biological parameters to variation in tributary inflow [7, 13–15]. At the same time, flow is at the epicenter of natural resource conflicts in the region because inflow to the SFE represents the water supply for 25 million people and irrigation water for a multi-billion dollar agricultural industry [16, 17].

A focal point for the debate over estuarine flows in the SFE is the Delta Smelt, an endangered fish endemic to the upper SFE [18, 19]. This osmerid is a slender-bodied planktivore typically reaching 60–70 mm standard length (SL) with a year-long life span. While the population employs a variety of life history strategies within the SFE [20], many Delta Smelt complete much of their life cycle in the Low Salinity Zone (LSZ; 0–6 psu region) of the upper estuary, and use the freshwater regions of the system primarily for spawning [19, 21]. Although abundance of Delta Smelt has been highly variable, there has been a long-term decline in abundance, including a notable step-change around 2002 [16, 22]. The decline of Delta Smelt has been intensively studied, with the general consensus that the decline has likely been caused by the interactive effects of multiple factors, including changes in physical and biotic components of habitat [16, 23–25]. One of the most controversial aspects of Delta Smelt management is the effect of flow on their habitat and abundance [21]. Specifically, physical habitat for Delta Smelt with suitable salinities is relatively large during high tributary inflow conditions when the LSZ is located downstream in the Suisun Bay region (Fig 1), but this area contracts during low inflow conditions when salinity intrusion pushes the LSZ upstream into the narrower tidal reaches of the Sacramento and San Joaquin Rivers [14, 15, 26]. For example, Suisun Bay and Suisun Marsh are thought to be one of the most important rearing areas for Delta Smelt [21,

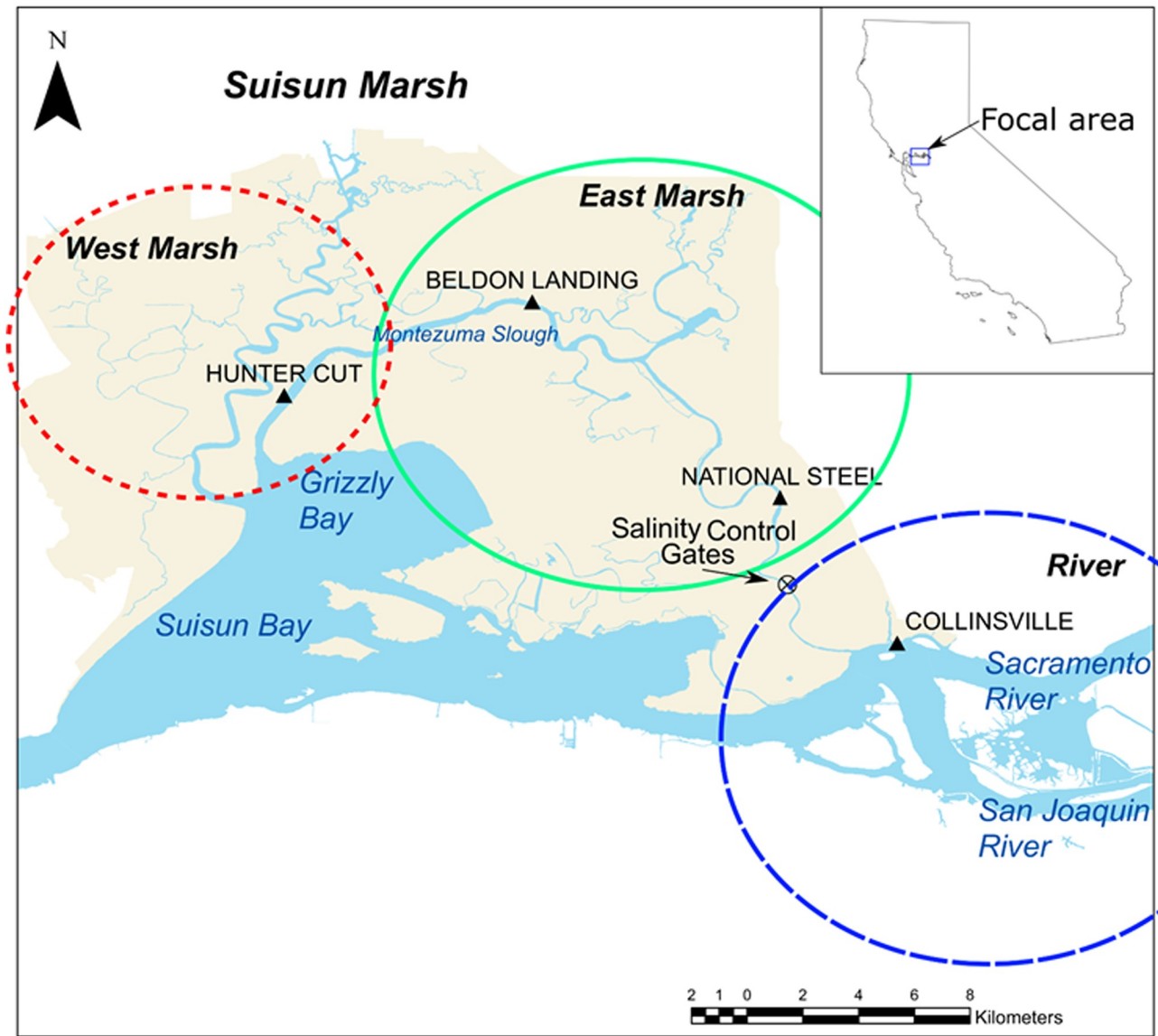

**Fig 1. The upper San Francisco Estuary, which includes Sacramento-San Joaquin Delta and Suisun Bay is located in the center of California, USA.** Each of the three primary sampling regions is circled in ovals: West Marsh, East Marsh, River.

27, 28]. Key attributes of this region include high turbidities, hydrodynamic complexity, more primary productivity, and extensive tidal wetlands. However, low inflows and associated salinity intrusion during late summer and fall commonly forces young smelt to move upstream out of the marsh and into less suitable habitat of the lower Sacramento and San Joaquin Rivers [15, 29]. The net effect is that the total habitat for the species tends to be much smaller in low inflow seasons such as summer and fall, and during all seasons in drought years. The relative importance of inflow in driving abundance therefore remains a source of substantial research interest [15], but a consistent pattern is that the abundance of the species is poor during drought conditions [18, 21].

Based on these findings related to inflow and companion data for other species, the current regulatory framework for the SFE includes mandated flows throughout various portions of the

ecosystem (ecosystem flows) during winter-spring [30], as well as additional inflow during fall of wetter years [31]. These ecosystem flows are accomplished within a remarkably complex infrastructure that distributes water to major agricultural regions and municipalities across large areas of California. This network includes upstream dams, managed floodways, large water diversions, and a system of canals and downstream reservoirs. Nonetheless, conflicts over water and the associated flows remain a core issue in California, leading to substantial interest in maximizing the ecosystem benefits of water management [21].

Here we describe a managed flow action ("Flow Action") designed to improve summer habitat conditions for Delta Smelt in Suisun Marsh, one of the large tidal marshes of the upper SFE (Fig 1). The marsh is located in between San Francisco Bay and the Delta, a network of tidal channels formed by the confluence of the Sacramento and San Joaquin Rivers. The Suisun Marsh region is located in a mixing zone between the eastern marine (San Francisco Bay) and western freshwater (Delta) regions. The general seasonal pattern is for salinity to increase in summer and fall in Suisun Marsh as freshwater inflow decreases. Temperature gradients vary throughout the year, but a general pattern is that air and water temperatures tend to be coolest toward the ocean (westward) in the winter and tend to be warmest inland (eastward) in summer. Bathymetry in Suisun Marsh and the adjacent Suisun and Grizzly Bays tends to be shallower than the heavily-channelized Delta. Shallower bathymetry promotes wind-wave resuspension of sediment and corresponding high turbidity during summer and fall [32].

We were able to target a Flow Action in the Suisun Marsh region of the SFE because of a unique engineered facility, the Suisun Marsh Salinity Control Gates (SMSCG; Fig 1). Details of how we used this facility to achieve a Flow Action are provided in the Methods. While the SMSCG facility and its operations were originally designed to decrease salinity for managed waterfowl habitat, we posited that directing low salinity water into the marsh would have substantial benefits for Delta Smelt during seasonal low tributary inflow periods when Suisun Marsh habitat would otherwise be too salty. The basic approach was to operate the SMSCG during the month of August, a critical time period for Delta Smelt rearing that is earlier than the facility is usually operated. Gate operations coincided with a managed tributary inflow augmentation of an estimated $45.6 \times 10^6$ m$^3$ from the Sacramento River, which was directed into Suisun Marsh using the SMSCG. We posited that: 1) directing low salinity water into Suisun Marsh would increase habitat area for Delta Smelt; 2) the benefits of the Flow Action would extend well beyond the August period of SMSCG operations; and 3) the resulting habitat conditions in Suisun Marsh would be superior to those in the lower Sacramento River. Specific hypotheses included:

- Operating the SMSCG in conjunction with an inflow augmentation would decrease salinity in Suisun Marsh to levels more suitable for Delta Smelt (<6 psu).

- Water quality variables (turbidity, temperature, chlorophyll, *Microcystis* blooms) would show seasonal patterns, but would not measurably change in response to the Flow Action.

- Following typical regional patterns, Suisun Marsh would have higher turbidity, chlorophyll, and zooplankton levels, and would have lower temperatures than the upstream Sacramento River.

- Reducing salinity in Suisun Marsh via the Flow Action would result in increased catch of Delta Smelt in that region.

More broadly, our goal was to quantify the estuarine ecosystem response to the Flow Action and evaluate the potential of such actions to support Delta Smelt and their habitat. To evaluate

these predictions, we compared field data collected in 2018 to historical data from other years, and we compared data collected in 2018 before, during, and after the Flow Action.

## Materials and methods

The authors of this study relied on data sources collected by multiple long-term agency surveys. These are covered under the Endangered Species Act through a permit to the Interagency Ecological Program, the umbrella organization for the work. Individual surveys were also covered by Scientific Collecting Permits issued by California Department of Fish and Wildlife.

### Flow action-Suisun Marsh Salinity Control Gates

The SMSCG facility was constructed in the late-1980s to counteract the effects of estuarine salinity intrusion into Suisun Marsh that resulted from upstream water diversions. It is located in the eastern portion of Montezuma Slough (Latitude 38.0934, Longitude -121.887) and was designed to tidally pump low salinity water into Suisun Marsh to maintain low salinities for wildlife habitat, primarily waterfowl. Montezuma Slough is a relatively broad open channel that supplies tidal water into the various reaches of Suisun Marsh. Under conditions when the SMSCG are not operating, on flood tide, relatively higher salinity water from Suisun Bay moves into Montezuma Slough at the western end of Montezuma Slough. On ebb tide, relatively lower salinity water from the lower Sacramento River moves into Montezuma Slough at the eastern end of Montezuma Slough. During summer conditions when the SMSCG are not operating, there is relatively little net flow through Montezuma Slough, even though the entire Montezuma Slough and Suisun Marsh area experiences flow reversal on a tidal time scale.

The SMSCG facility consists of three radial gates, a maintenance channel, and a boat lock (Fig 2). The gates are intermittently or continuously operated between October and May to maintain regulatory standards for salinity. Under SMSCG operations, the radial gates are lowered during the flood tides and opened during the ebb tides (i.e. tidally operated). The SMSCG control salinity by directing low salinity water from the Sacramento River into Montezuma Slough during ebb (outgoing) tides but restricting the tidal flow of higher salinity water into Montezuma Slough during flood (incoming) tides. This strategy generates a net flow of low salinity water from east to west in Montezuma Slough of approximately 63 m$^3$/s, which is considerably larger than the net flow during the summer when the gates are not operating. These operations result in fresher water in Suisun Marsh, as well as the northern part of Grizzly Bay. Note that directing Sacramento River inflow into Suisun Marsh via the SMSCG results in some upstream (eastward) salinity intrusion in Suisun Bay from San Francisco Bay; hence, operation of SMSCG often coincides with additional water releases from dams and increased Sacramento River inflow into the SFE (or alternatively decreased water diversions) to offset this intrusion, in order to meet regulated salinity standards in the interior Delta. Another way to describe this pairing of an inflow augmentation with the use of existing infrastructure is that additional flows are released from upstream tributary reservoirs combined with the SMSCG being used to direct additional low salinity water into Suisun Marsh. We call this combined inflow augmentation and SMSCG operation throughout the paper the "Flow Action", to capture the pairing of two actions to achieve low-salinity habitat for Delta Smelt in Suisun Marsh. As noted in the Introduction, gate operations coincided with a managed tributary inflow augmentation of an estimated 45.6 x 10$^6$ m$^3$ from the Sacramento River, which was directed into Suisun Marsh using the SMSCG.

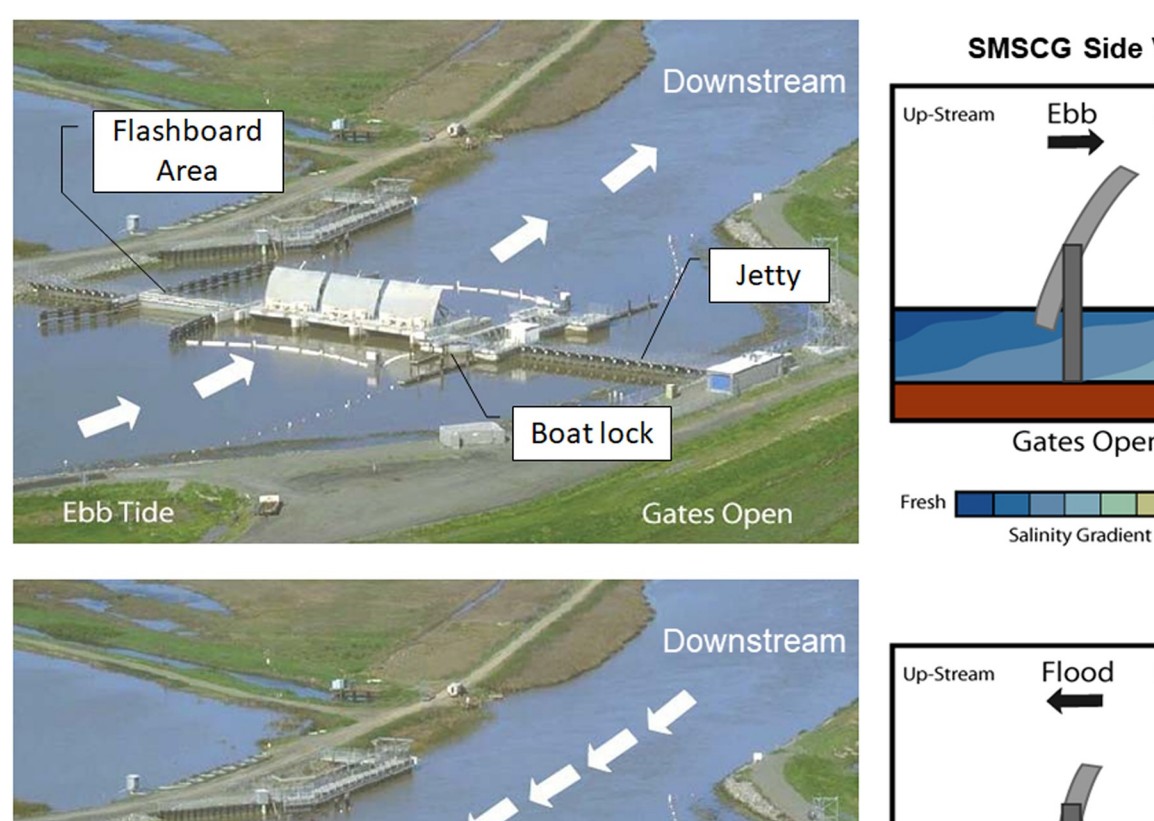

**Fig 2. Suisun Marsh Salinity Control Gates.** Under typical operations, the gates are opened during ebb tide (upper panels) and closed during flood tide (lower panels) to tidally pump low salinity water into Suisun Marsh.

### Study approach

To evaluate the effects of the Flow Action on Delta Smelt habitat, we conducted a field monitoring program spanning before, during, and after the Flow Action. This monitoring plan was informed in part by predictions from the application of a three-dimensional (3D) hydrodynamic model of the entire San Francisco Bay and Sacramento-San Joaquin Delta that was conducted before the Flow Action to predict potential effects on Delta Smelt habitat [33]. This modeling work, in addition to previously-developed conceptual models [34, 35], provided the biological framework and spatiotemporal scope for hypothesis testing in this study. We used the same hydrodynamic model in analyses to compare modeled 2018 conditions to modeled conditions if the Flow Action had not occurred, to directly estimate the effect of the Flow Action on a subset of physical components of the habitat. Although the study included fish sampling to assess the response of Delta Smelt to the action, we focused on examining habitat

variables because Delta Smelt catch is so rare that it is difficult to directly measure their response.

To address the major study hypotheses, our design included a spatial and temporal component with environmental measurements before and after the Flow Action, as well as comparisons between the habitat in Suisun Marsh ("Marsh") and the upstream region of the Sacramento River ("River"). The habitat evaluations included discrete and continuous field measurements of habitat conditions, as well as three-dimensional modeling of spatiotemporal changes in the LSZ area.

## Data sources

**Flow.** Daily flow data estimating the net freshwater outflow from the Delta to the SFE were accessed at the DWR website DAYFLOW https://data.cnra.ca.gov/dataset/dayflow. DAY-FLOW is a water balance model that estimates the daily average outflow from the Delta. The DAYFLOW output, known as the Net Delta Outflow Index (NDOI), is used for many regulatory and operational purposes. While DAYFLOW uses a complex formula to estimate NDOI —reflecting the complexity of water supply infrastructure in the Delta—it can be roughly interpreted as the difference between Delta inflows and water exports. DAYFLOW does not account for tidal flows or the large-scale estuary drain-fill cycles driven by the lunar Spring-Neap tidal cycle, and NDOI therefore can differ significantly from other approaches such as an estimated water balance using field estimates of cross-sectional water flows.

Water flow through Suisun Marsh (through Montezuma Slough) was computed from Acoustic Doppler Current Profiler (ADCP) measurements of water velocity at National Steel (Fig 1) using the index velocity method [36]. The total net flow diverted into Montezuma by action was computed as the cumulative summation of the instantaneous flows in Montezuma Slough over the duration of the action.

**Water quality.** Continuous water quality measurements were taken using Yellow Springs International (YSI) EXO2 multi-parameter water quality probes at three locations (Fig 1). These sites were selected because they spanned a broad range of the study area: "River" (Sacramento River at Collinsville;"East Marsh" (National Steel); and"West Marsh" (Hunter's Cut). The probes were used to measure temperature, turbidity, specific conductivity (as a surrogate for salinity), and chlorophyll fluorescence at 15-minute intervals during the July-October period. Note that each probe sampled water from relatively large areas of each region because of strong tidal flows through the study area. For example, we estimate that typical tidal flows are approximately +/-5700 m$^3$/s in the Sacramento River near Collinsville and +/- 1800 m$^3$/s at the mouth of Montezuma Slough near Hunter's Cut.

**Food web sampling.** Chlorophyll fluorescence was measured during the study using the previously-described water quality sensors. Zooplankton and *Microcystis* sampling relied on data from several long-term boat-based sampling programs: the Interagency Ecological Program's Environmental Monitoring Program (EMP) (https://emp.baydeltalive.com/projects/11285), which collects zooplankton samples on a monthly basis at fixed sites throughout the upper SFE [37]; California Department of Fish and Wildlife's (CDFW) Summer Townet Survey (STN) (https://www.wildlife.ca.gov/Conservation/Delta/Townet-Survey), which collects zooplankton and fish samples at fixed sites on a biweekly basis in July and August; and the Fall Midwater Trawl (FMWT) (https://www.wildlife.ca.gov/Conservation/Delta/Fall-Midwater-Trawl), which operates on a monthly basis in September-December and also collects zooplankton samples in addition to fish sampling at fixed sites in the study area (Fig 1; S1 Table). Sampling methods were similar between surveys. For each zooplankton sample, an 85 cm long, 12.5 cm diameter zooplankton net made of 0.16 mm nylon mesh was attached to a steel

sampling sled and towed through the water in a stepwise oblique manner for ten minutes to sample the entire water column. A General Oceanics model 2030 flowmeter was fixed in the mouth of the net to calculate water volume sampled. Samples were preserved in 10% formalin and subsamples were identified by taxonomists at CDFW's laboratory in Stockton, CA. Catch per cubic meter of water sampled (CPUE) was calculated as:

$$N = \frac{C}{(M_s - M_e)kAS}$$

Where:
$N$ = the number of a taxon per cubic meter of water filtered
$C$ = the cumulative number of a taxon counted
$M_s$ = starting flowmeter reading
$M_e$ = ending flowmeter reading
$k$ = flowmeter constant
$A$ = net mouth area
$S$ = fraction of total sample examined

CPUE was converted to biomass per cubic meter (BPUE) by using published conversion factors [38–40].

*Microcystis* presence was assessed visually by survey personnel concurrently with zooplankton and fish sampling. At each sampling event, a bucket of water was collected and presence of *Microcystis* colonies was ranked on a scale of 1–5, with 1 being "absent" and 5 being "very high". These data were converted to "presence" or "absence" for analysis.

**Fish sampling.**   Delta Smelt abundance and distribution data were collected by U.S. Fish and Wildlife Service as part of the Enhanced Delta Smelt Monitoring Program, a year-round effort initiated in November 2016 [41]. The basic approach was to sample eight different strata of the upper SFE with a Generalized Random Tessellation Stratified design [42] using multiple tows at each location, with each survey lasting one week. We focused on the July-October period, which targets sub-adult Delta Smelt. The sample gear consisted of a Kodiak trawl towed by a pair of vessels [43]. The Kodiak trawl had maximum mouth dimensions of 1.83 m x 7.62 m, with a body consisting of five panels of stretch mesh starting at 5.08-cm near net mouth and decreasing to 0.64 cm at cod end, which was attached to a live car. During each survey period a Kodiak trawl sampled 3–4 sites per day with a minimum of 2 tows per site. If no Delta Smelt were captured in the first two tows, one or two additional samples were taken. Depending on conditions, this approach allowed 24–36 sites to be sampled each survey period. For each tow, the total number of Delta Smelt was counted, measured, and individuals were saved for future research. The results for each weekly sampling period and region were summarized based on catch per trawl (CPUE).

Since Delta Smelt have become so rare, we also examined historical catch data for Suisun Marsh based on the Summer Townet Survey, described above. We focused on data from the past two decades before the Flow Action (2002–2017) since that period best corresponds to the current ecological regime [16, 44] and because this was the same period that we evaluated for hydrology and water quality.

## Data analyses

Our analytical approach used a combination of descriptive and quantitative statistics, as well as three-dimensional hydrodynamic simulation modeling. The hydrodynamic modeling was used to simulate conditions throughout the San Francisco Bay and Sacramento-San Joaquin Delta both with the Flow Action and without the Flow Action. Comparing the results of these

hydrodynamic modeling simulations allowed for direct estimates of the effects of the Flow Action on conditions in Montezuma Slough and Suisun Marsh, either at individual locations through time or as spatial maps.

Comparisons with historical water quality data were based on visual comparisons of the data, and a statistical analysis for salinity and temperature (see below). We did not evaluate historical data for chlorophyll, turbidity, zooplankton, or fish because comparable samples were not collected in previous years. For historical comparisons, the mean monthly value for each water quality parameter was calculated for the same stations and years. We then plotted the mean and standard error of the daily means for the chosen dry years, wet years, and 2018, described below.

To provide a historical perspective for our analysis, we examined how 2018 compared to recent "dry" and "wet" years. We first did a hierarchical cluster analysis for 2003–2017 to identify the major hydrological groupings based on July salinity. For the highest salinity (driest) grouping, there were three years (2002, 2009, 2016) with very similar summer salinity conditions. We did not include two recent dry years (2014 and 2015) because they occurred during a historic drought, when the estuary had much more salinity intrusion than 2018. Similarly, we used the wettest grouping to select three high flow "wet" summers (2005, 2006, 2017). These general "dry" and "wet" years were used as the basis for comparing 2018 environmental variables to historical conditions.

For the historical statistical analysis, we calculated the mean of salinity and temperature collected at National Steel (the site where the effect was hypothesized to be greatest) and modeled the daily mean with normal additive error structures using the formula:

$$Mean = Yeartype \text{ x } Month + lag1 + lag2$$

Where

*Mean* = Daily mean of parameter value.

*Yeartype* = Categorical variable indicating whether data were from the action year (2018), the historical dry summers (2002, 2009, 2016), or the historical wet summers (2005, 2006, 2017)

*Month* = July, August or September. Included as a factor to test differences before (Jul), during (Aug), or after (Sep) the action

*lag1* = A one-day lag in the value of "Mean"

*lag2* = A two-day lag in the value of "Mean"

The lag values were included to account for autocorrelation after examining the autocorrelation and partial autocorrelation function plots for each parameter. Visual assessments of parameter autocorrelation were verified by fitting autoregressive moving average (ARMA) models to each parameter and assessing the significance of each model term. Including the one and two-day lag captured all significant autocorrelation effects in the environmental parameters modeled here. Data were also checked for assumptions of normality and homogeneity of variance.

To evaluate differences in the observed data for environmental parameters before, during, and after the action, as well as differences between the River, West Marsh, and East Marsh sites, we used a second series of linear models on the data from 2018 with additive normal error structures. All models were checked for assumptions of normality and homogeneity of variance, and all models were checked for spatial and temporal autocorrelation.

For continuous water quality parameters, we calculated the daily mean for each water quality parameter (turbidity, chlorophyll, temperature, and salinity) collected at each site and

modeled the daily mean using the formula:

$$Mean = Station \text{ x } Month + lag1 + lag2$$

Where

*Station* = Eastern Marsh (National Steel), Western Marsh (Hunters Cut), or Sacramento River (Collinsville)

Other variables are the same as for the historical data model, above.

For *Microcystis* levels and zooplankton samples, which were collected at several locations twice per month, the model was

$$Value = Region + Month$$

Where

*Value* = log-transformed BPUE for zooplankton and presence/absence for *Microcystis*.

*Region* = Either Suisun Marsh (samples collected within Montezuma Slough) or River (Samples collected near the confluence of the Sacramento and San Joaquin Rivers). See Fig 1 for sampling locations.

Note that there were insufficient samples of zooplankton and *Microcystis* to subdivide the Suisun Marsh region into West Marsh and East Marsh regions, so a single geographic grouping was used. Since *Microcystis* was presence/absence instead of a continuous value, we used a binomial generalized linear model. We assessed both models with and without a Region x Month interaction term, and in both cases there was no improvement in AIC for the model with the interaction term, so we used the less complex model. All models and associated analyses were conducted using R version 3.6.1 (2019, R Foundation for Statistical Computing). Because we only have the complete data set for a single year, we cannot conclusively say that the Flow Action, and not seasonal changes, caused the resulting trends. However, these models can be discussed in regards to our expectations for seasonal and gate-action-related changes.

To visually assess how the suite of water quality parameters changed during the summer of 2018, we used non-metric multi-dimensional scaling (NMDS) of the Manhattan dissimilarity index of the daily average of the continuous probe data at each site (salinity, temperature, chlorophyll, and turbidity) [45, 46]. NMDS is a non-parametric ordination method that collapses information from multiple dimensions and graphically displays them in two dimensions such that the distance between points on the plot approximates the pairwise dissimilarities between each point as closely as possible [45]. We plotted the NMDS scores for each sample with vectors of each water quality parameter. We then used a permutational multivariance analysis of variance (PERMANOVA) to test whether the three regions (East Marsh, West Marsh, and River) and three time periods (July, August, and September) were significantly different in water quality. We conducted post-hoc pairwise comparisons of region and month and used a Bonferroni correction on the p-values to account for multiple tests. These analyses were done using the vegan package in R [47].

**Habitat modeling.** A key tool in our evaluations was the use of a 3D hydrodynamic model to simulate spatial and temporal patterns in physical habitat with and without the Flow Action. Initial hydrodynamic model simulations examining the effects of a hypothetical Flow Action under historic conditions were conducted prior to the field scale implementation and were used in the initial planning for the 2018 efforts, informing some of the predictions for field measurements of environmental variables [33].

The UnTRIM Bay-Delta model is a three-dimensional hydrodynamic model of San Francisco Bay and the Sacramento-San Joaquin Delta [48] developed using the UnTRIM hydrodynamic model [49, 50]. The UnTRIM Bay-Delta model extends from the Pacific Ocean through

the entire Sacramento-San Joaquin Delta. Tidal and non-tidal water levels, salinity, and water temperature are applied at the Pacific Ocean open boundary. The UnTRIM Bay-Delta model includes spatially varying meteorology, freshwater inflow from tributaries to the Bay and Delta, and water exports/intakes throughout the system. Water control structures, such as gates and temporary barriers, are also included in the model. The UnTRIM Bay-Delta model has been calibrated using water level, water flow, and salinity data collected in the SFE in numerous previous studies [14, 48] and was validated for the 2018 Flow Action period using observed water level, salinity, and temperature data collected in the study area.

Because the hydrodynamic model domain is significantly larger than the study area in Suisun Marsh and both the upstream flows and ocean water levels are unaffected by the Flow Action, all boundary conditions remained identical for the historical simulation of the Flow Action and the corresponding scenario that did not include the Flow Action, such that only the SMSCG gate operation and Delta exports specified in the model varied between the scenarios. In the hydrodynamic model, the SMSCG were operated following the same schedule and operational triggers used by the California Department of Water Resources to operate the physical facility. The SMSCG are represented by three grid cells in the model: one representing flashboards, one representing the radial gates, and one representing the boat lock and jetty on the eastern side of Montezuma Slough. The width of each grid cell corresponds to the width of each SMSCG component. Flow through the boat lock is represented using a rating curve. During the period when the flashboards are in place, water flow is not allowed through the water grid cell representing the flashboards.

Simulated salinity and temperature with and without the Flow Action in the flow area were then summarized for the periods before, during, and after the Flow Action. Hydrodynamic model results from the simulation with the Flow Action were compared to results from the simulation without the Flow Action, to estimate the effects of the Flow Action on salinity and temperature. Time series were extracted from the model to evaluate salinity and temperature through time at discrete locations. Depth-averaged salinity and temperature at each model grid cell were time-averaged for the period of the Flow Action and the period following the Flow Action to create spatial maps and visualize how the effects of the Flow Action vary in space. The percent of time the depth-averaged salinity was less than 6 psu at each model grid cell was also calculated for both simulations, and to visualize the change resulting from the Flow Action. This approach allowed us to evaluate the effects of the Flow Action on salinity and temperature over the entire study area rather than just at the locations of the water quality probes.

As a specific example of how the modeling data were used, we summarized salinity patterns at a central Suisun Marsh location (Beldens Landing; Fig 1) with and without the Flow Action, allowing us to identify the magnitude and duration of the effect. These data were also used to examine how salinity changes in Suisun Marsh could have influenced habitat suitability for Delta Smelt. Specifically, we first plotted modeled August 2018 salinity distributions with and without the Flow action. Modeled salinity changes due to the Flow Action were then compared to historical responses of Delta Smelt to Suisun Marsh salinity. For the latter analysis, Delta Smelt response to salinity was modeled using a binomial distribution based on presence/absence in the CDFW Summer Townet Survey in Suisun Marsh during August of 2002–2017. We converted electrical conductivity measured by the Summer Townet Survey to salinity using the 'ec2pss' function from the 'wql' package in Program R [51].

## Results

Below we summarize the Results based on several data sources. The data presented include: 1) historical Suisun Marsh and the River in years when there was no Flow Action; 2) observed

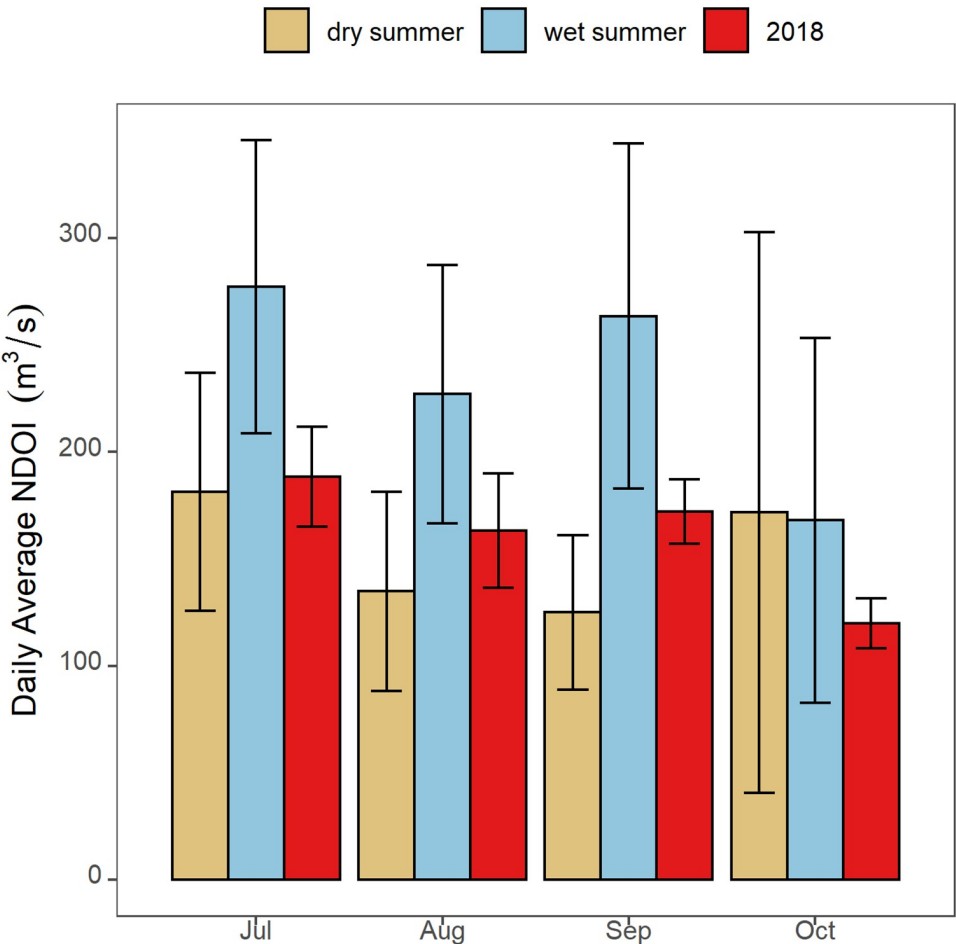

**Fig 3. Net Delta Outflow (NDOI) for 2018 (red bars) as compared to previous wet (blue bars) and dry (brown bars) summers.** Error bars represent standard errors of the daily mean.

responses during the 2018 Flow Action; and 3) simulations of the SFE using a hydrodynamic model.

## Flow

In summer and fall 2018 NDOI ranged from 113 to 231 $m^3$/s, representative of typical seasonal changes, water operations and water quality regulations (Fig 3). NDOI through the SFE was comparable to earlier years with dry summer and fall conditions. The SMSCG were operated August 2 –September 7, diverting an estimated 196 x $10^6$ $m^3$/s of low salinity water through Suisun Marsh. To maintain compliance with regional water quality standards and avoid salinity intrusion into the estuary, total net Delta outflow was augmented with an estimated inflow pulse of 45.6 x $10^6$ $m^3$ during the SMSCG operation period. The flow augmentation required to maintain compliance was estimated to be 28.5% of the total flow diverted through Suisun Marsh.

## Water quality

In all months, the general pattern was that there was significantly higher salinity in Suisun Marsh than in the upstream River (Fig 4; Table 1). In July, before the Flow Action, salinity in

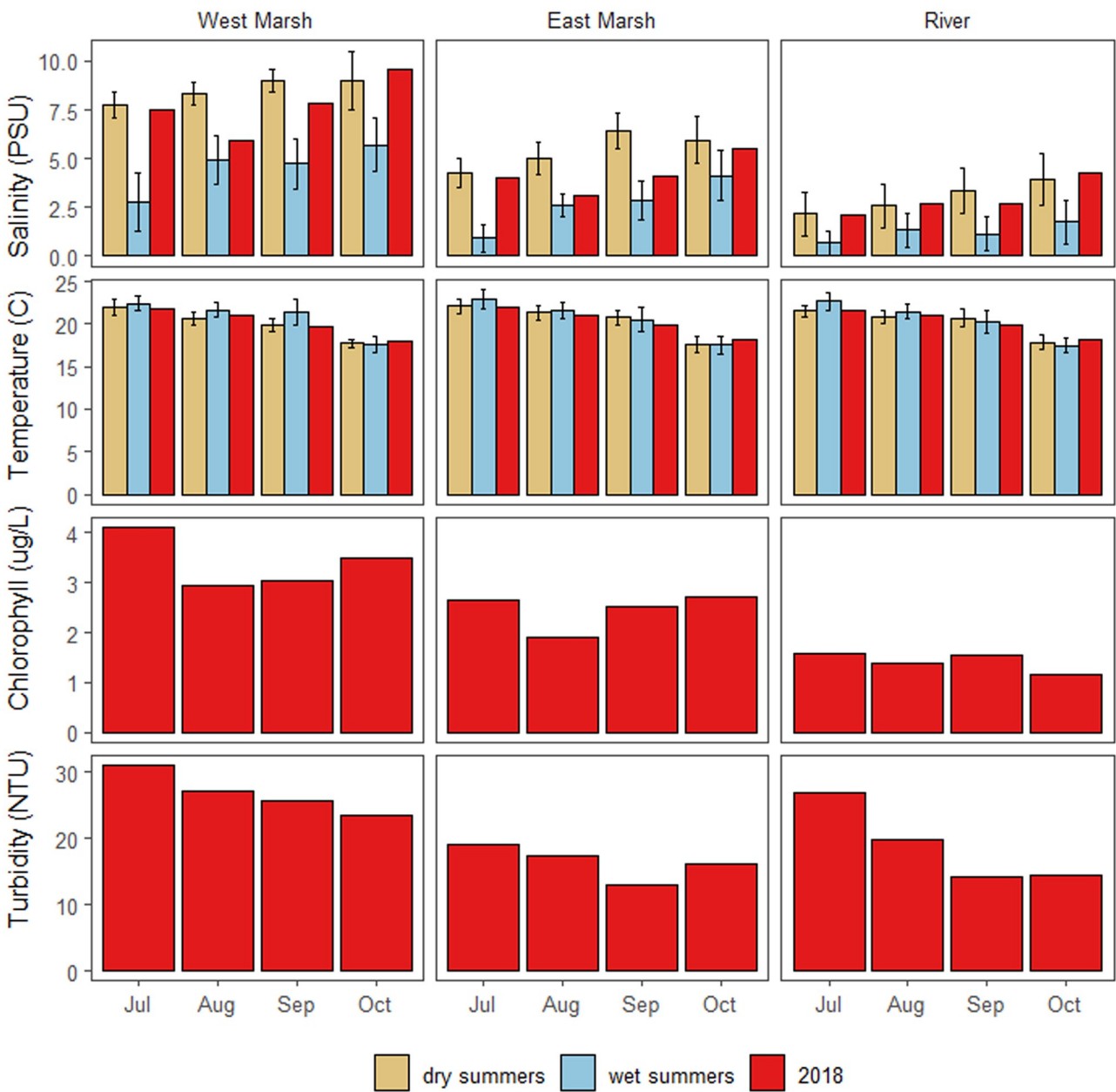

**Fig 4. Monthly water quality results for three continuous monitoring locations: Sacramento River ("River"), East Suisun Marsh ("East Marsh") and West Suisun Marsh ("West Marsh").** The 2018 results (red bars) are shown in addition to historical dry (brown bars: 2002; 2009; 2012) and wet (blue bars: 2005; 2006; 2017) summers. Error bars represent standard errors of the daily mean. Statistical differences between years and months for the East Marsh are listed in Table 1.

the River and two Marsh locations ("East Marsh" and "West Marsh") was comparable to historical dry summers, with significantly higher salinities than historical wet summers (Fig 4, Table 1). As expected, the major effect of the Flow Action was to significantly reduce salinities in Suisun Marsh during August as compared to July (Fig 4; Table 1). The net effect was that August salinity conditions were comparable to historical wet years in both the East and West Marsh areas(Fig 4). By contrast, there was no substantial change in River salinity during the Flow Action compared to historical dry summers. Marsh salinity conditions in August were

**Table 1. Linear model results for salinity and temperature for the action year (2018) versus historical dry (2002; 2009; 2012) and wet (2005; 2006; 2017) summers.**

| Salinity (PSU)–Adjusted $R^2$: 0.921 | | | | | |
|---|---|---|---|---|---|
| Null deviance: 2014.45 on 709 degrees of freedom | | | | | |
| Residual deviance: 156.12 on 699 degrees of freedom | | | | | |
| Predictor | Estimate | St. Error | t-value | p-value | |
| Intercept: July, action year | 1.755 | 0.114 | 15.378 | <0.0001 | *** |
| Month: August | -0.463 | 0.087 | -5.306 | <0.0001 | *** |
| Month: September | 0.105 | 0.086 | 1.220 | 0.223 | |
| Historic wet summer | -1.428 | 0.105 | -13.645 | <0.0001 | *** |
| Historic dry summer | 0.093 | 0.079 | 1.167 | 0.244 | |
| One-day lag | 0.644 | 0.035 | 18.354 | <0.0001 | *** |
| Two-day lag | -0.087 | 0.031 | -2.760 | 0.006 | ** |
| Aug X wet summer | 1.283 | 0.122 | 10.490 | <0.0001 | *** |
| Sep X wet summer | 0.773 | 0.120 | 6.432 | <0.0001 | *** |
| Aug X dry summer | 0.838 | 0.116 | 7.209 | <0.0001 | *** |
| Sep X dry summer | 0.910 | 0.123 | 7.403 | <0.0001 | *** |
| Temperature (degrees C)–Adjusted R2: 0.768 | | | | | |
| Null deviance: 967.99 on 606 degrees of freedom | | | | | |
| Residual deviance: 220.94 on 596 degrees of freedom | | | | | |
| Predictor | Estimate | St. Error | t-value | p-value | |
| Intercept: July, action year | 9.294 | 0.495 | 18.790 | <0.0001 | *** |
| Month: August | -0.623 | 0.110 | -5.668 | <0.0001 | *** |
| Month: September | -1.174 | 0.115 | -10.174 | <0.0001 | *** |
| Historic wet summer | 0.344 | 0.103 | 3.332 | 0.001 | *** |
| Historic dry summer | -0.098 | 0.114 | -0.862 | 0.389 | |
| One-day lag | 0.582 | 0.037 | 15.776 | <0.0001 | *** |
| Two-day lag | 0.004 | 0.033 | 0.108 | 0.914 | |
| Aug X wet summer | -0.036 | 0.142 | -0.253 | 0.800 | |
| Sep X wet summer | -0.006 | 0.146 | -0.039 | 0.969 | |
| Aug X dry summer | 0.215 | 0.160 | 1.344 | 0.180 | |
| Sep X dry summer | 0.602 | 0.162 | 3.710 | 0.000 | *** |

fresher and similar to historical wet years (Fig 4), and this pattern continued through September in the East Marsh. The model comparing salinity in the East Marsh to historic years showed a significant interaction of month and year type, with 2018 being similar to historic dry years in July, but closer to historic wet years in August and September (Fig 4). By October, salinity conditions in both Marsh regions were again similar to historical dry summers, as they were in July before the Flow Action began.

The UnTRIM Bay-Delta model predictions provide additional insight into the salinity effects of the combined effects of inflow augmentation and SMSCG operation. The simulation of August 2018 showed that Marsh habitat conditions were suitable (<6 psu salinity) for Delta Smelt a much higher percentage of time with the Flow Action (Fig 5a) than if we had not performed the Flow Action (Fig 5b). The net change in habitat suitability was apparent for many of the Marsh channels, and extended into the broad shoals of Grizzly Bay (Fig 5c). The modeling results also suggest that the habitat benefits of reduced salinities from the Flow Action extended more than a month past the period when the SMSCG were operated for the Flow Action (August 2-September 7). For example, simulation of salinities at Belden Landing, a central marsh location that helps to illustrate general regional patterns, showed that marsh salinities under the Flow Action were substantially lower than the No-Action simulation for a

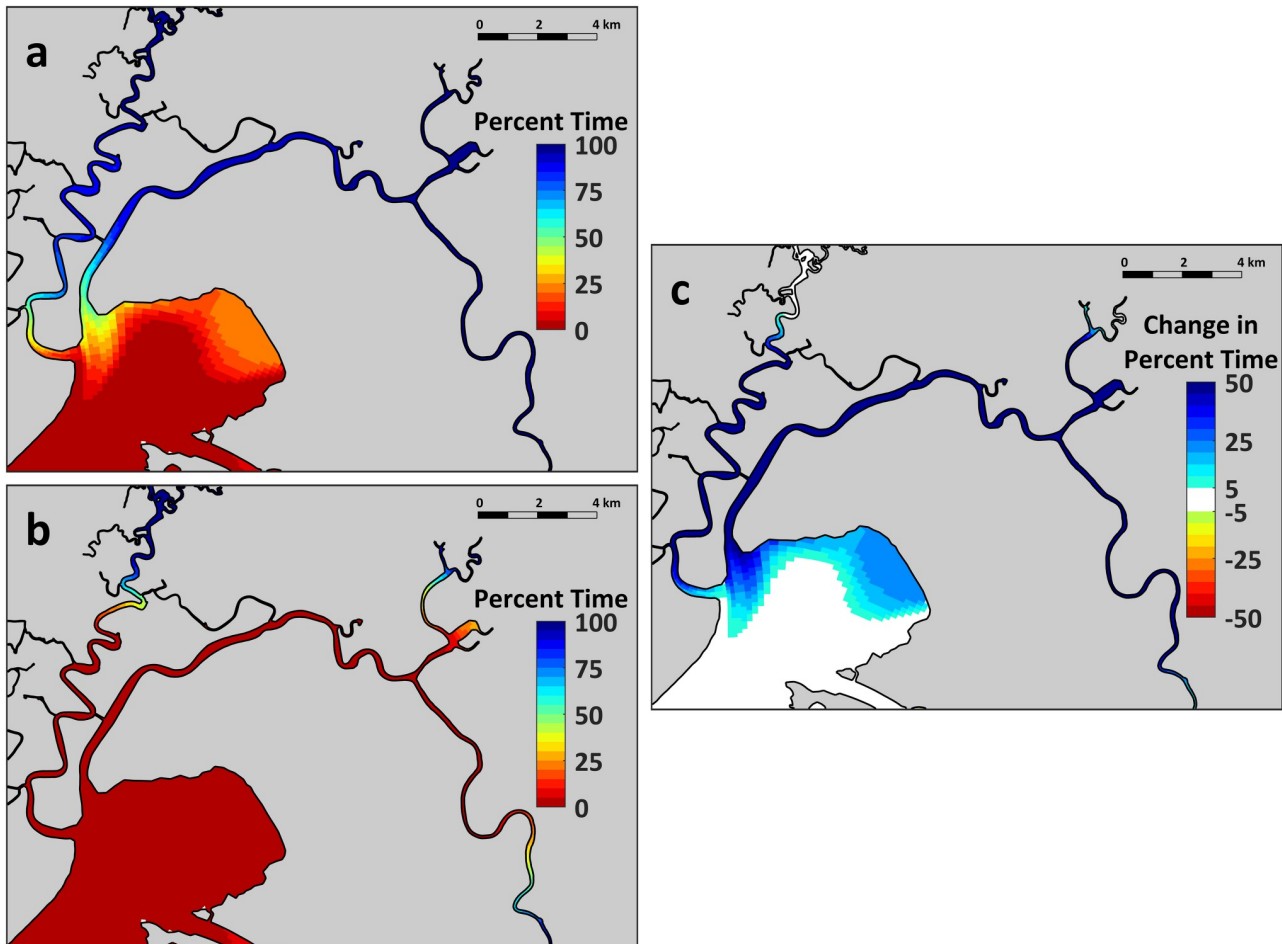

**Fig 5. UnTRIM 3-D model simulation of salinity distribution during August 2018 for the Suisun region: a: Flow Action; b: No Flow Action; c: Net difference between the two simulations.** The results are summarized based on the percentage of the time that salinities were below 6 psu, a widely-used upper threshold to represent low salinity habitat in the SFE [7] and to represent the habitat of Delta Smelt [27, 52].

month and a half beyond the end of the gate operation period (Fig 6). The salinity effects of the Flow Action therefore appear to subside to near zero by mid-October, when typical SMSCG operations often occur to support waterfowl habitat, frequently resulting in a seasonal decrease in salinity. While Marsh salinities in September and October 2018 were comparable to those measured in historical dry years (Fig 4, Table 1), the modeling results show that Marsh salinities would have been higher than the measured values in these months if the Flow Action had not taken place.

Temperature followed the expected seasonal pattern, with cooler temperatures in September compared to the July and August summer months (Fig 4; Table 2). These monthly temperatures were comparable to historical dry and wet summers, with no apparent effect of the Flow Action. The model comparing temperature in the East Marsh to historic years showed that historic wet summers were usually slightly warmer than 2018, and that dry summers in September were also slightly warmer than expected (Table 1). The simulation modeling predicted only minimal effects of the Flow Action on temperature. It is therefore not surprising that the Flow Action did not detectably alter temperatures in either the Marsh or the River. Contrary

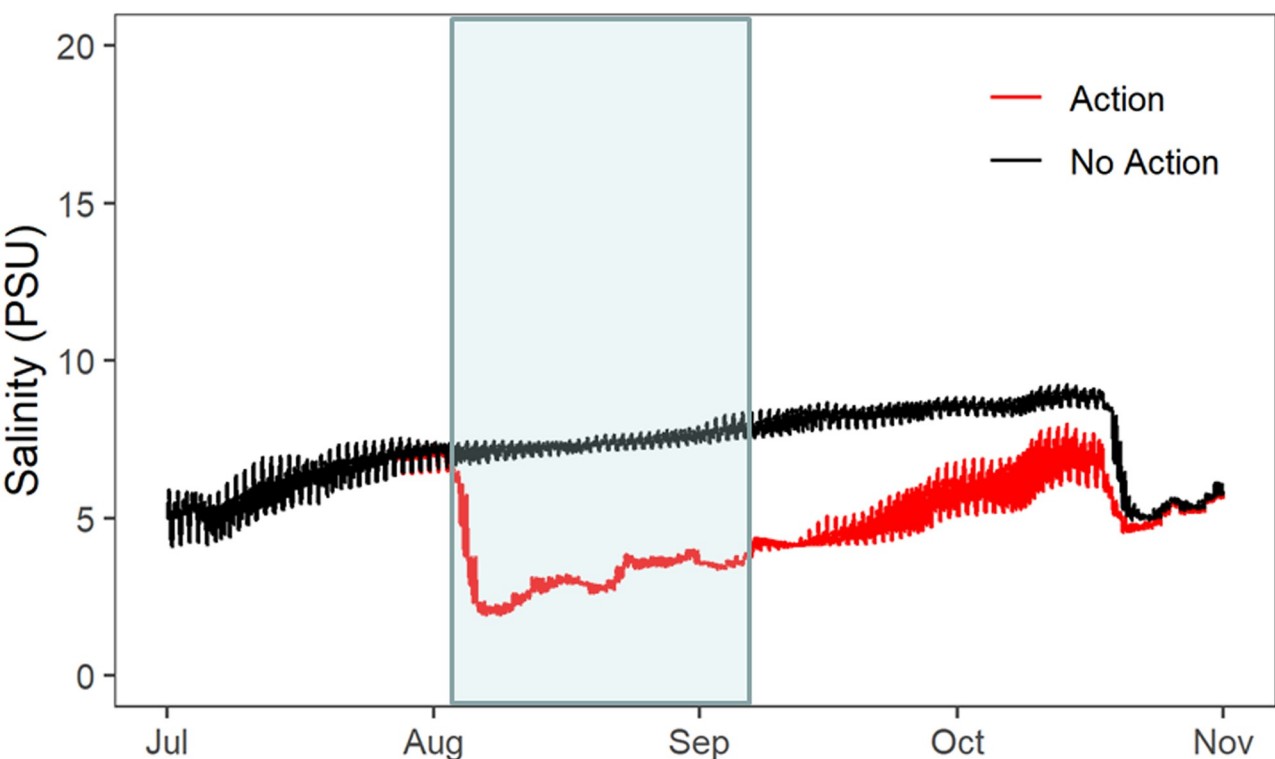

**Fig 6. UnTRIM 3-D modeled salinity (psu) in the eastern Marsh (Belden's Landing) for simulations with and without the Flow Action.** The period of gate operations is highlighted in blue.

to our predictions, temperatures in the two Marsh sites were not significantly different than the upstream River.

Observed turbidity at discrete locations showed a strong seasonal pattern, with significantly lower turbidity levels in later months (Fig 4; Table 2). There was nothing in the data to suggest that the Flow Action in August deviated from this seasonal progression. Turbidity decreased less in the Western Marsh than in the River in late summer, indicating that there were regional differences. Unlike turbidity, chlorophyll did not show a consistent seasonal pattern (Fig 4; Table 2). However, chlorophyll was significantly higher in the two Marsh locations than in the River during all months.

The suite of continuous water quality parameters (above) was examined using an NMDS approach, which allowed us to look at the overall changes due to season, location, and the Flow Action (Fig 7). Water quality varied substantially across the three regions, reflecting the general change in water quality in moving downstream from tidal river channels into more brackish marsh habitat. Monthly patterns were also apparent at each monitoring site, with a notable shift in water quality during the August Flow Action, followed by an additional major shift during September. The PERMANOVA analysis of the NMDS results indicated that the set of water quality variables was significantly different before, during and after the action, and different between the three regions (Table 3).

## Food web

Zooplankton biomass at all sites was dominated by copepods, principally *Acartiella* and *Pseudodiaptomus* (Fig 8). *Tortanus* was rare in in the Sacramento River sites, but was present each

**Table 2. Linear model results for water quality and lower trophic data for the action month (August) versus before and after the Flow Action and between regions for 2018.**

Salinity (PSU)–Adjusted $R^2$: 0.99

Null deviance: 2758.617 on 455 degrees of freedom

Residual deviance: 30.121 on 442 degrees of freedom

| Predictor | Estimate | St. Error | t-value | p-value | |
|---|---|---|---|---|---|
| Intercept: July, River | 0.330 | 0.061 | 5.448 | <0.0001 | *** |
| Month: August | 0.131 | 0.055 | 2.372 | 0.018 | * |
| Month: September | 0.120 | 0.056 | 2.142 | 0.033 | * |
| Station: West Marsh | 0.984 | 0.142 | 6.946 | <0.0001 | *** |
| Station: East Marsh | 0.349 | 0.070 | 4.981 | <0.0001 | *** |
| One-day lag | 1.294 | 0.054 | 24.026 | <0.0001 | *** |
| Two-day lag | -0.461 | 0.052 | -8.935 | <0.0001 | *** |
| Aug X West Marsh | -0.466 | 0.094 | -4.967 | <0.0001 | *** |
| Sep X West Marsh | -0.091 | 0.079 | -1.155 | 0.249 | |
| Aug X East Marsh | -0.315 | 0.083 | -3.796 | <0.0001 | *** |
| Sep X East Marsh | -0.086 | 0.078 | -1.110 | 0.268 | |

Turbidity (NTU, log-transformed)—Adjusted $R^2$: 0.9361

Null deviance: 56.2059 on 455 degrees of freedom

Residual deviance: 4.9394 on 442 degrees of freedom

| Predictor | Estimate | St. Error | t-value | p-value | |
|---|---|---|---|---|---|
| Intercept: July, River | 0.532 | 0.090 | 5.934 | <0.0001 | *** |
| Month: August | -0.061 | 0.025 | -2.485 | 0.014 | * |
| Month: September | -0.108 | 0.029 | -3.669 | 0.000 | *** |
| Station: West Marsh | 0.013 | 0.024 | 0.561 | 0.575 | |
| Station: East Marsh | -0.066 | 0.025 | -2.589 | 0.010 | * |
| One-day lag | 1.321 | 0.056 | 23.659 | <0.0001 | *** |
| Two-day lag | -0.483 | 0.057 | -8.513 | <0.0001 | *** |
| Aug X West Marsh | 0.038 | 0.033 | 1.155 | 0.249 | |
| Sep X West Marsh | 0.076 | 0.035 | 2.168 | 0.031 | * |
| Aug X East Marsh | 0.051 | 0.033 | 1.523 | 0.129 | |
| Sep X East Marsh | 0.058 | 0.034 | 1.727 | 0.085 | . |

Temperature (degrees C)–adjusted $R^2$: 0.97

Null deviance: 1263.457 on 455 degrees of freedom

Residual deviance: 33.851 on 442 degrees of freedom

| Predictor | Estimate | St. Error | t-value | p-value | |
|---|---|---|---|---|---|
| Intercept: July, River | 2.249 | 0.427 | 5.267 | <0.0001 | *** |
| Month: August | -0.081 | 0.040 | -2.039 | 0.042 | * |
| Month: September | -0.204 | 0.052 | -3.920 | <0.0001 | *** |
| Station: West Marsh | 0.017 | 0.039 | 0.427 | 0.670 | |
| Station: East Marsh | 0.037 | 0.040 | 0.937 | 0.350 | |
| One-day lag | 1.429 | 0.050 | 28.729 | <0.0001 | *** |
| Two-day lag | -0.532 | 0.052 | -10.310 | <0.0001 | *** |
| Aug X West Marsh | -0.012 | 0.054 | -0.216 | 0.830 | |
| Sep X West Marsh | -0.038 | 0.055 | -0.684 | 0.494 | |
| Aug X East Marsh | -0.034 | 0.055 | -0.632 | 0.528 | |
| Sep X East Marsh | -0.025 | 0.055 | -0.456 | 0.649 | |

Chlorophyll Fluorescence (µg/L, log-transformed)—Adjusted $R^2$: 0.9612

Null deviance: 91.9810 on 455 degrees of freedom

*(Continued)*

**Table 2.** (Continued)

Residual deviance: 4.8501 on 442 degrees of freedom

| Predictor | Estimate | St. Error | t-value | p-value | |
|---|---|---|---|---|---|
| Intercept: July, River | 0.048 | 0.018 | 2.642 | 0.009 | ** |
| Month: August | -0.006 | 0.021 | -0.286 | 0.775 | |
| Month: September | -0.001 | 0.021 | -0.048 | 0.962 | |
| Station: West Marsh | 0.140 | 0.029 | 4.755 | <0.0001 | *** |
| Station: East Marsh | 0.064 | 0.024 | 2.650 | 0.009 | ** |
| One-day lag | 1.252 | 0.056 | 22.303 | <0.0001 | *** |
| Two-day lag | -0.373 | 0.056 | -6.620 | <0.0001 | *** |
| Aug X West Marsh | -0.075 | 0.030 | -2.528 | 0.012 | * |
| Sep X West Marsh | -0.049 | 0.031 | -1.596 | 0.112 | |
| Aug X East Marsh | -0.032 | 0.030 | -1.083 | 0.280 | |
| Sep X East Marsh | 0.003 | 0.030 | 0.115 | 0.908 | |

Zooplankton BPUE (µgC/m3)—Adjusted R2: 0.2314

Null deviance: 9.824 on 39 degrees of freedom

Residual deviance: 6.970 on 36 degrees of freedom

| Predictor | Estimate | St. Error | t-value | p-value | |
|---|---|---|---|---|---|
| Intercept: July, River | 8.259 | 0.143 | 57.589 | <0.0001 | *** |
| Month: August | -0.335 | 0.170 | -1.978 | 0.056 | . |
| Month: September | 0.003 | 0.173 | 0.020 | 0.985 | |
| Region: Suisun Marsh | -0.419 | 0.140 | -2.992 | 0.005 | ** |

Microcystis—binomial presence/absence

Null deviance: 63.449 on 50 degrees of freedom

Residual deviance: 42.088 on 48 degrees of freedom

| Predictor | Estimate | St. Error | t-value | p-value | |
|---|---|---|---|---|---|
| Intercept: July, Confluence | 1.350 | 0.833 | 1.624 | 0.105 | |
| Month: August | -0.083 | 0.986 | -0.084 | 0.934 | |
| Month: September | -0.979 | 1.048 | -0.934 | 0.351 | |
| Month: October | -19.949 | 2634.50 | -0.008 | 0.994 | |
| Region: Suisun Marsh | -2.892 | 0.8616 | -3.363 | 0.001 | *** |

month in Suisun Marsh. There were no significant differences in biomass between July, August, and September (Fig 8; Table 2). However, Suisun Marsh zooplankton biomass densities were significantly lower on compared to the upstream Sacramento River.

Microcystis colonies (based on visual assessment) remained at relatively low levels throughout the sampling period (Fig 9). However, Suisun Marsh had significantly lower levels than the upstream Sacramento River (Table 2), suggesting slightly better water quality conditions in the tidal marsh habitat, at least with respect to harmful algal blooms such as *Microcystis*.

## Delta Smelt

Delta Smelt catch was rare throughout the study, reflecting the dire status of this imperiled species. During July-October a total of only 136 Delta Smelt were caught, and caught only in four regions of the SFE: Suisun Bay, Suisun Marsh, Lower Sacramento River, and the Sacramento Deep Water Ship Channel (Fig 10). The catch patterns for Delta Smelt were qualitatively consistent with our overall prediction that the Flow Action would allow Delta Smelt to colonize Suisun Marsh, but we cannot make claims as to overall population effects of the action with

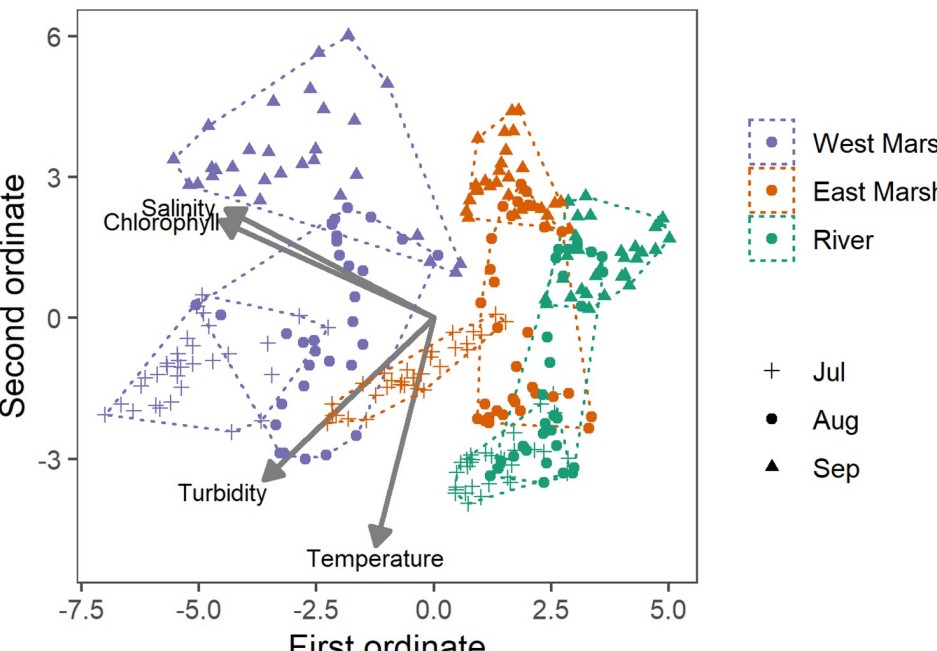

**Fig 7. NMDS multivariate results for the three continuous water quality stations: River; East Marsh; and West Marsh.** The results are grouped by month to show the seasonal changes. As noted in the text, the flow pulse action occurred primarily in August.

any confidence. Seven Delta Smelt were caught in Suisun Marsh during the Flow Action but none were detected in Suisun Marsh before or after the action.

Since Delta Smelt were so rare in 2018, we also examined how the species historically responded to salinity in Suisun Marsh based on the Summer Townet Survey. To help interpret the responses in 2018, we overlaid modeled salinity data with and without the Flow Action (see above). The historical catch data illustrate that salinity has a strong effect on the presence of Delta Smelt in Suisun Marsh (Fig 11A). Moreover, the Flow Action resulted in salinities that correspond to a higher historical probability of Delta Smelt presence in Suisun Marsh (Fig 11B).

**Table 3. Results of a Permutational Multivariate Analysis of Variance (PERMANOVA) on the salinity, temperature, chlorophyll, and turbidity for the River, East Marsh and West Marsh regions between months (July, August, September).** (corresponding to groups in NMDS plot).

| Overall PERMANOVA | | | | | |
|---|---|---|---|---|---|
| | **Degrees of Freedom** | **Sums of Squares** | **F-Value** | **R²** | **p-value** |
| Region | 2 | 0.5471 | 1117.95 | 0.659 | 0.001 |
| Month | 2 | 0.2172 | 443.92 | 0.262 | 0.001 |
| Residuals | 271 | 0.0663 | | 0.080 | |
| Pairwise tests | | | | | |
| Jul. v. Aug. | 1 | 0.0686 | 32.98 | 0.139 | 0.003 |
| Jul. v. Sept | 1 | 0.1933 | 76.78 | 0.316 | 0.003 |
| Aug. v. Sept | 1 | 0.0751 | 34.33 | 0.164 | 0.003 |
| East Marsh v. West Marsh | 1 | 0.2891 | 267.32 | 0.595 | 0.003 |
| West Marsh v. River | 1 | 0.4734 | 434.67 | 0.7049 | 0.003 |
| East Marsh v. River | 1 | 0.0581 | 61.500 | 0.2526 | 0.003 |

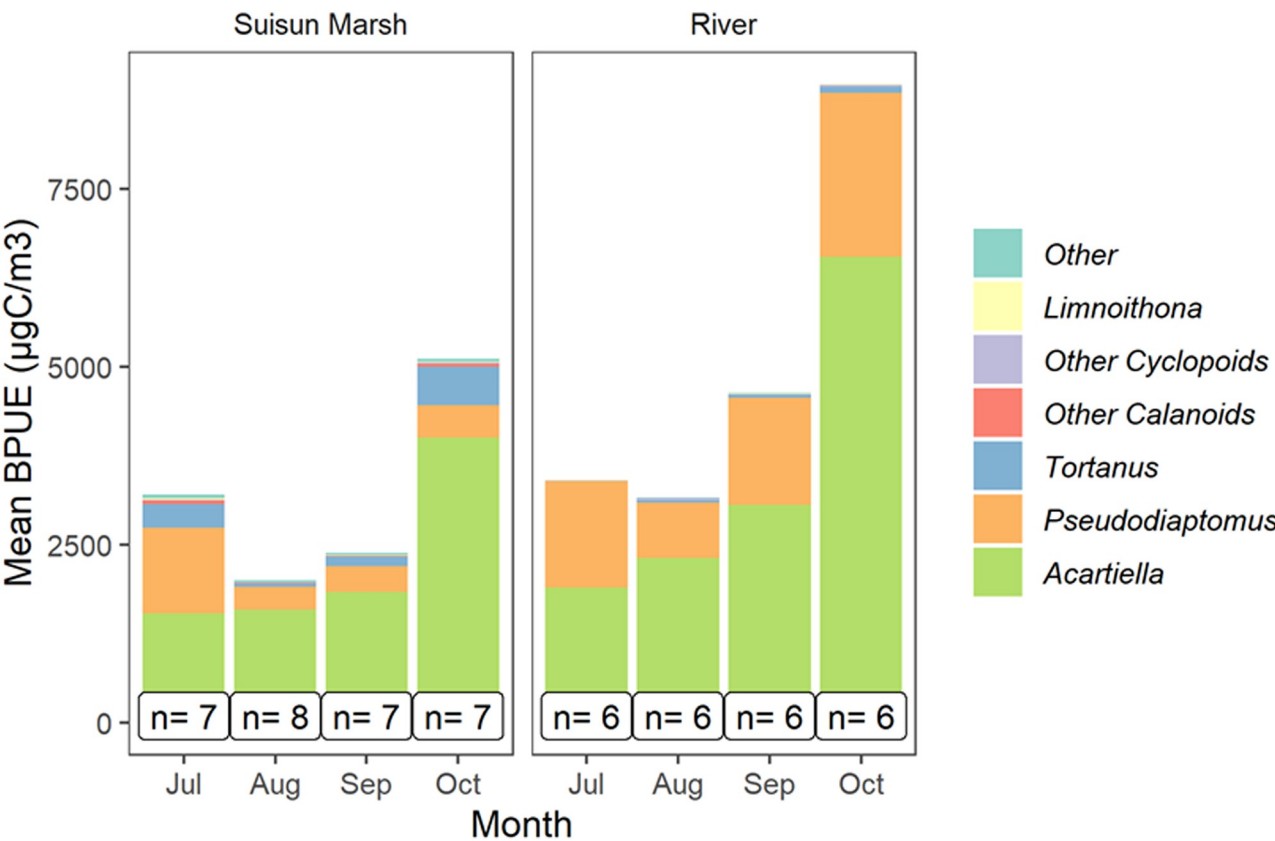

**Fig 8. Mean monthly biomass per unit effort of zooplankton (ug C/m3) for the River and Marsh regions.** The taxonomic groups that comprised each monthly summary are illustrated with different colors. Sample sizes are provided for each month during 2018.

## Discussion

The environmental effects of tributary inputs on estuaries is a well-understood concept in coastal and estuarine ecology. The most obvious influence of variability in tributary inputs is the shifting location of the salinity field [53, 54], with additional effects from upstream transport of organic and inorganic material, as well as redistribution of organisms, particularly those that are sensitive to changes in salinity [13, 55–57].

Our study allowed us to intensively look at ecological effects across a key region of one of the largest estuaries on the Pacific Coast. Large-scale manipulations of physical habitat are rare and laden with challenges in managing the timing, magnitude, and spatial distribution of tributary inputs to estuaries [1]. In the case of the Flow Action, we were able to generate a flow pulse using a unique tidal gate facility that helped focus the response into the target region. Since effects of inputs are so complex, we evaluated the responses of a large tidal wetland complex at multiple ecological scales. Below we summarize some of the major physical, chemical and biological responses, with emphasis on changes resulting from the Flow Action, and how habitat conditions differed across geographical regions and years. Overall, we believe that the ecological effects of the action were substantial, and provide insight into the management value of freshwater flow inputs to estuaries.

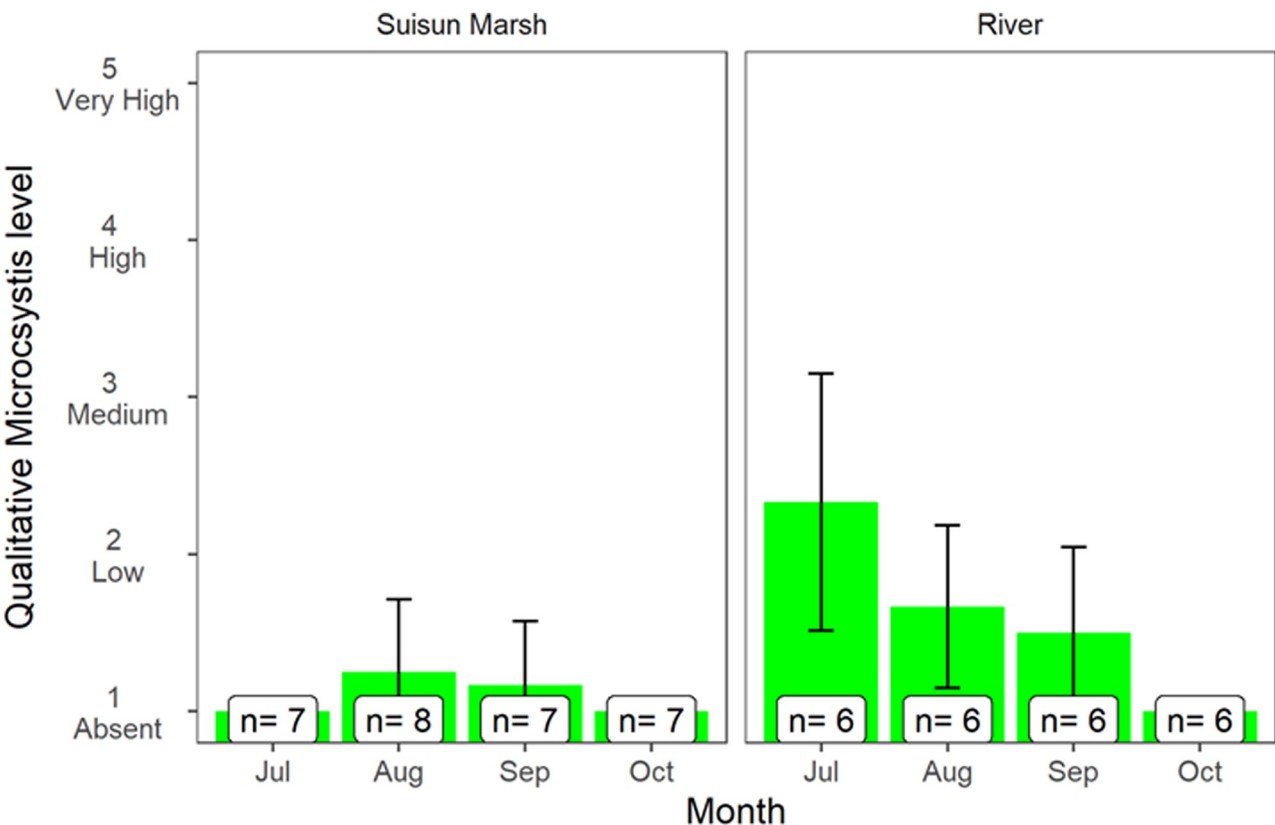

**Fig 9. Presence of *Microcystis* colonies for monthly observations during 2018 in the River and Marsh regions.** Sample sizes and standard deviations are provided for each month.

## Physical and chemical responses

As expected, the major response of the management action was to decrease salinities across Suisun Marsh, the largest contiguous marsh in the SFE. Using continuous water quality sensors and hydrodynamic modeling, we found that the action was able to reduce salinity across the axis of Suisun Marsh, and also the open water shoal habitat of Grizzly Bay (Fig 5). This is despite the fact that the volume of low salinity water directed through Suisun Marsh (196 x 10$^6$ m$^3$) during the 36-day Flow Action represents only about 9.6 hours of peak tidal flows in the Sacramento River (i.e. +/-5700 m$^3$/s). The broad geographic footprint of the Flow Action (Fig 5) is consistent with the design of the SMSCG facility, which relatively efficiently directs freshwater flow through the Marsh corridor and its seaward outlet at Grizzly Bay. Another notable finding from examination of the modeled data was that the salinity effects of the management action persisted for weeks after the cessation of the Flow Action. This suggests that there was a substantial lag before tidal forcing was able to disperse low salinity water directed into the Marsh during the Flow Action. This type of lag in the response of the salt field to fresh water flow is not unusual as delayed changes in salinity commonly occur in other estuaries because of complex hydrodynamic and geographic interactions [58].

For both the upstream River region and Suisun Marsh, there was no evidence that the action substantially changed some of the other key water quality parameters such as temperature and turbidity. Although levels of these habitat constituents varied across months consistent with long-term seasonal patterns (Fig 4), temperature and turbidity did not show clear

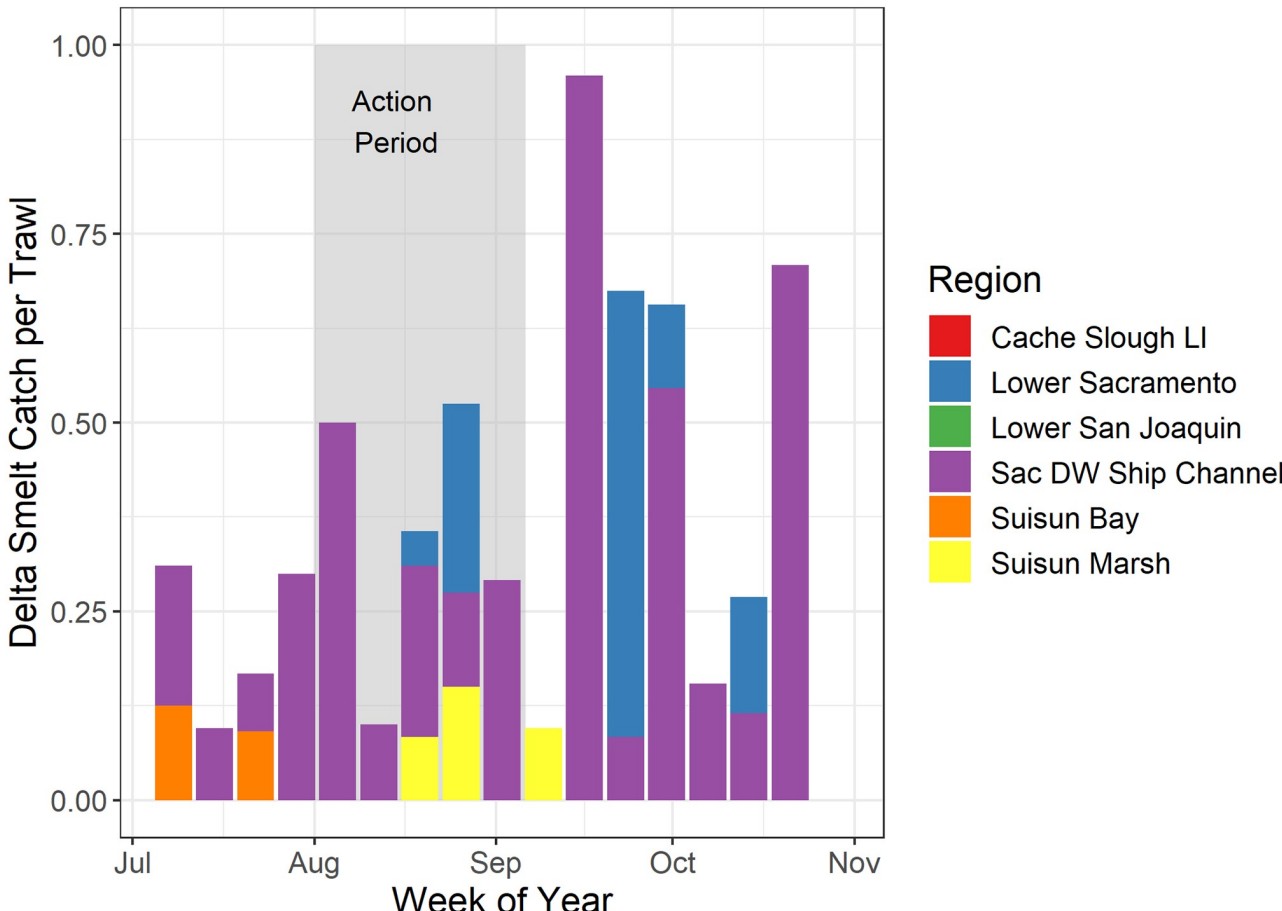

**Fig 10. Catch-per-unit-effort (CPUE–catch/tow) of Delta Smelt across six different regions of the upper estuary during summer and fall 2018.**
The period of the Flow Action is highlighted. Details about the survey and locations are summarized in USFWS, 2019 [41].

changes coincident with the Flow Action. This finding is not surprising since seasonal water temperatures in this region are largely controlled by air temperature [59], and dry-season (i.e. summer and early fall) turbidities are more a function of wind wave resuspension than tributary sediment inputs [32, 60, 61]. This does not mean that the management action did not have ecological effects with respect to either of these Delta Smelt habitat variables. The Flow Action moved the low salinity zone seaward into Suisun Marsh and Grizzly Bay, which have shallow water habitat that is subject to wind-wave resuspension. These regional differences are apparent based on the higher turbidity levels in Suisun Marsh than in the upstream Sacramento River channel, which is relatively narrow, deep, and channelized. Therefore, the net result of the management action was that the dynamic low salinity habitat was present in a higher turbidity zone, improving overall habitat conditions for the target species Delta Smelt (Fig 5), which thrive in turbid conditions [15].

Unlike turbidity, we did not observe any major differences in water temperature between the Suisun Marsh and upstream River channels. This was unexpected as SFE shows a general trend with cooler air and water temperatures in seaward locations during the summer and early fall [59]. We had therefore predicted slightly cooler water temperatures in Suisun Marsh than the upstream Sacramento River channel during the study period, and that lowering salinities during the SMSCG action would provide Delta Smelt with access to lower temperature

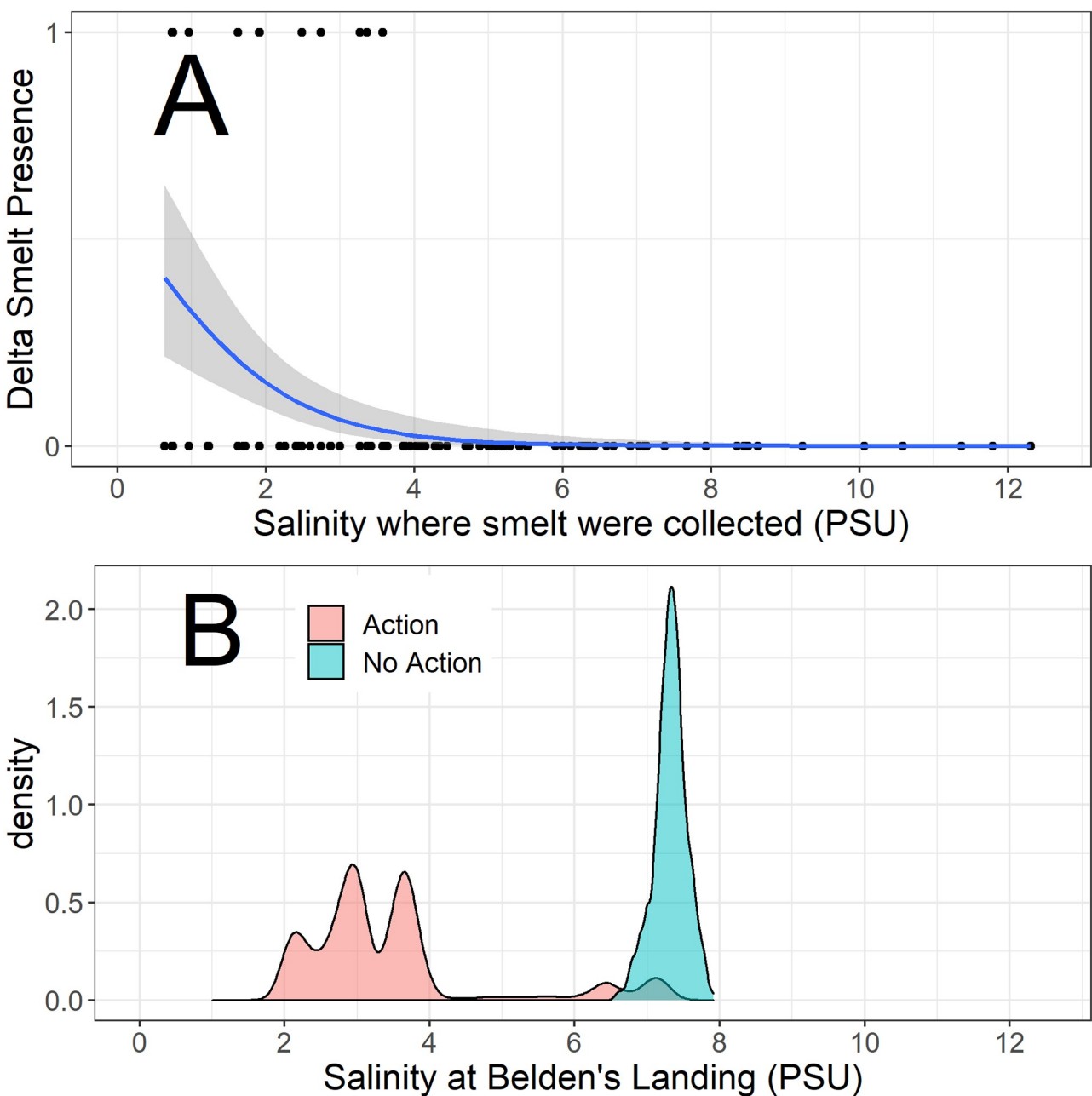

**Fig 11.** A: Comparison of historical (2002–2017) August Delta Smelt TNS catch data for Suisun Marsh in relation to measured salinities during the surveys. A binomial model was fitted to develop the relationship, (p < 0.0001). B: Distributions of modeled August 2018 salinities with and without the Flow Action. The modeled salinities are for Beldens Landing, a location central to Suisun Marsh and TNS sampling.

conditions. One possible reason that Marsh channels were not cooler than the River is that many of its channels penetrate relatively far inland, where air temperatures tend to be warmer during summer. Alternatively, it is possible that our River temperature measurements were not sufficiently landward to detect the expected pattern. In any case, like other estuaries, temperature is a major concern as many native species such as Delta Smelt are relatively sensitive to high temperatures [62], so it is important to understand if there are regional temperature refuges that could help the species cope with heat waves and climate change.

## Biological responses

The management action did not result in a detectable change in lower trophic levels. Although there was a drop in chlorophyll at both the Marsh and River locations during August, inspection of the continuous water quality data indicates that these declines began before the August 2 start date of the Flow Action. We therefore do not believe that the Flow Action was the main driver of overall chlorophyll patterns during August. As predicted, chlorophyll was at higher levels in Suisun Marsh than the Sacramento River throughout the study. This difference in primary production is likely because of longer residence time of water and hydrodynamic complexity in the Marsh habitat as compared to the Sacramento River [63]. Reducing salinity in Suisun Marsh therefore provided Delta Smelt access to a large habitat area with higher phytoplankton biomass, a beneficial outcome considering the relatively low productivity of the San Francisco estuary [64, 65].

It is also important to note that *Microcystis* remained at low levels before, during, and after the management action, indicating that food web responses were not confounded by harmful algal blooms. This is notable when considering that the management action coincided with typical seasonal peaks in Microcystis abundance—an increasing water quality problem, especially in the Delta and in below normal inflow years such as 2018 [66]. It is possible that the Flow Action may have helped reduce levels of *Microcystis* since its blooms tend to be reduced by higher outflow conditions [67].

Contrary to our prediction, zooplankton levels were somewhat lower in Suisun Marsh than the Sacramento River, despite higher average phytoplankton concentration (as indicated by chlorophyll) in the marsh habitat. This is consistent with previous research showing lower zooplankton biomass in brackish habitat (> 0.55 psu [28]) as compared to upstream lower salinity areas. For copepods, the primary prey of Delta Smelt, high grazing pressure from introduced predatory zooplankton and bivalves may work to limit the responses to higher phytoplankton biomass [68, 69].

Due to the extreme rarity of Delta Smelt, we cannot conclusively say that the Flow Action allowed these fish to colonize the Suisun Marsh Habitat. However, Delta Smelt were caught in the marsh during the Action and were not caught before or after the action. These general patterns matched our expectations that the species would not occupy the marsh until salinities were reduced during the Flow Action. This observation is also consistent with our analysis of historical Delta Smelt catch data for Suisun Marsh, which indicated that the Flow Action resulted in salinities with a much greater probability of Smelt presence in the marsh (Fig 11). Like many other estuarine species, the distribution of Delta Smelt shows a relatively strong relationship with salinity [15, 70], so the apparent response of limited numbers of fish and the historical modeling results were reasonable.

Previous studies suggest that Delta Smelt that reside in Suisun Marsh have improved nutritional and growth indices compared to individuals in other geographic areas [71]. Moreover, the general goal of increasing distribution of this species is consistent with its recovery plan [72], and with population theory that suggests that improved distribution and life history diversity can help reduce risks of localized catastrophic events [73]. On the other hand, the population remains highly imperiled and this action alone was not sufficient to generate a detectable change in population size, nor do we expect that a single action like this would be sufficient to facilitate recovery of the species even with multiple years of operations.

## Management implications

While the broader effects of flow in estuaries have been well described [7, 13], our study provides insight into the effects of localized and event-based inputs. Although estuaries are often

dominated by tidal flows, our investigation suggests that manipulating tidal flows, via the SMSCG, in conjunction with a modest freshwater inflow pulse, can result in measurable changes to the salinity field, resulting in improved habitat suitability for an ESA-listed fish over a relatively large spatial area. The changes in salinity we observed persisted for several weeks beyond the completion of the management action. Collectively, these results highlight the potential management value of similar actions assuming the actions are crafted appropriately for target species or habitat areas. Note, however, we are not advocating construction of similar tidal gate facilities to achieve ecological goals for estuaries. In this case, the gates had already been constructed to meet Suisun Marsh water quality objectives for waterfowl and were readily available to test for flow effects. Moreover, we are not aware of any other candidate locations in the Delta where new gate structure could have similar benefits.

One outcome of our project is that the SMSCG system is being considered as part of regional long-term habitat management for the upper SFE to support Delta Smelt. Consistent with the goal of our study, the SMSCG operations during dry seasons are now included in conjunction with restoration as tools to improve access and habitat quality for this endangered species [74]. For example, ambitious restoration plans are underway to restore 3,200 ha of tidal wetland habitat across the Bay-Delta including Suisun Marsh [31]; https://water.ca.gov/Programs/All-Programs/EcoRestore). Use of the SMSCG in conjunction with freshwater flow pulses could help enhance the suitability of this area for Delta Smelt and other native species.

## Supporting information

**S1 Table. Sampling locations for zooplankton, *Microcystis*, fish, and water quality.** (DOCX)

## Acknowledgments

This project was developed as part of the Delta Smelt Resiliency Strategy, a multi-agency effort to improve the population status of this imperiled species. The effort was conducted with the guidance of the Interagency Ecological Program and the Collaborative Adaptive Management Team. We owe particular thanks to Denise Barnard and Catherine Johnson (USFWS), who led the EDSM Delta Smelt sampling effort. Additional helpful input was provided by the project team including: Jennifer Pierre (State Water Contractors); Shawn Acuna (Metropolitan Water District); John Durand and Ted Grosholz (UC Davis); Corey Graham (USFWS); Eli Ateljevich, Nicky Sandhu, Ian Ueker, Tiffany Brown, Betsy Wells, Ted Swift, Rhiannon Klingonsmith and Karen Gehrts (California Department of Water Resources). The initial concept for the flow action was suggested by David Fullerton (Metropolitan Water District) and Bill Bennett (Bay Institute). Permitting support was provided by Brooke Jacobs and Carl Wilcox (California Department of Fish and Wildlife) and Kaylee Allen (USFWS).

## Author Contributions

**Conceptualization:** Ted Sommer, Michal Koller, Michael Koohafkan, J. Louise Conrad, April Hennessy, Michael Beakes.

**Data curation:** Rosemary Hartman, Michael Koohafkan, Michael MacWilliams, Aaron Bever, Christina Burdi, April Hennessy.

**Formal analysis:** Ted Sommer, Rosemary Hartman, Michael Koohafkan, Michael MacWilliams, Aaron Bever.

**Funding acquisition:** Ted Sommer, J. Louise Conrad.

**Investigation:** Ted Sommer, Michael Koohafkan, J. Louise Conrad, Christina Burdi, Michael Beakes.

**Methodology:** Ted Sommer, Rosemary Hartman, Michael Koohafkan, J. Louise Conrad, Aaron Bever, Christina Burdi.

**Project administration:** Ted Sommer, Michal Koller, J. Louise Conrad.

**Resources:** Ted Sommer, Rosemary Hartman, Michal Koller, J. Louise Conrad.

**Supervision:** Ted Sommer, Michal Koller, April Hennessy.

**Validation:** Rosemary Hartman, Michael Koohafkan, Aaron Bever, April Hennessy.

**Visualization:** Rosemary Hartman, Michael Koohafkan, J. Louise Conrad, Michael MacWilliams, Aaron Bever, Christina Burdi.

**Writing – original draft:** Ted Sommer, Michael Koohafkan, Michael MacWilliams, Aaron Bever, Michael Beakes.

**Writing – review & editing:** Ted Sommer, Rosemary Hartman, Michal Koller, Michael Koohafkan, J. Louise Conrad, Michael MacWilliams, Aaron Bever, Christina Burdi, Michael Beakes.

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
