## [Decision Letter · Decision Letter 0]

24 Jun 2020

PONE-D-20-16291

Evaluation of a large-scale flow manipulation to the upper San Francisco Estuary: Response of habitat conditions for an endangered native fish

PLOS ONE

Dear Dr. Sommer,

Thank you for submitting your manuscript to PLOS ONE. After careful consideration, we feel that it has merit but does not fully meet PLOS ONE’s publication criteria as it currently stands. Therefore, we invite you to submit a revised version of the manuscript that addresses the points raised during by both reviewers during the review process. 

We look forward to receiving your revised manuscript.

Kind regards,

Vanesa Magar, Ph.D.

Academic Editor

PLOS ONE

Journal Requirements:

2) In your Methods section, please provide additional location information of the study site, including geographic coordinates for the data set if available.

3)  Thank you for stating the following in the Competing Interests section:

[The authors have declared that no competing interests exist.].   

We note that one or more of the authors are employed by a commercial company: Anchor QEA LLC

i) Please provide an amended Funding Statement declaring this commercial affiliation, as well as a statement regarding the Role of Funders in your study. If the funding organization did not play a role in the study design, data collection and analysis, decision to publish, or preparation of the manuscript and only provided financial support in the form of authors' salaries and/or research materials, please review your statements relating to the author contributions, and ensure you have specifically and accurately indicated the role(s) that these authors had in your study. You can update author roles in the Author Contributions section of the online submission form.

ii) Please also provide an updated Competing Interests Statement declaring this commercial affiliation along with any other relevant declarations relating to employment, consultancy, patents, products in development, or marketed products, etc. 

Reviewers' comments:

Reviewer's Responses to Questions

**Comments to the Author**

1. Is the manuscript technically sound, and do the data support the conclusions?

Reviewer #1: Partly

Reviewer #2: Yes

2. Has the statistical analysis been performed appropriately and rigorously? 

Reviewer #1: Yes

Reviewer #2: Yes

3. Have the authors made all data underlying the findings in their manuscript fully available?

Reviewer #1: Yes

Reviewer #2: No

4. Is the manuscript presented in an intelligible fashion and written in standard English?

Reviewer #1: Yes

Reviewer #2: Yes

5. Review Comments to the Author

Reviewer #1: Specific Comments

(1) Line 69. Can more detail be presented in Figure 1 such as the water course, on which the Salinity Control Gates are situated? The labels appear fuzzy to me, but perhaps that is due to the figure being converted to pdf format. This map is really important and should have sufficient detail for understanding the study. Why not start your map quadrat at Carquinez Strait so that you can provide greater detail? If you want, put an insert in the corner with San Francisco Bay. Yet, is it really important to show San Francisco Bay?

(2) Line 72. Why not start the Introduction here and place the proceeding information at the beginning of the Introduction?

(3) Line 114. Could this paragraph be moved into the Methods section? Again, I am trying to shorten the introduction so that the reader does not lose interest due to its length.

(4) Line 138. This is a very illustrative figure. It gives the reader a good idea of how salinity is regulated. Again, it would be better placed in the Methods.

(5) Line 142. Start up again with this paragraph. You would have to add some text to briefly describe the SMSCG here, but describe it in detail in the Methods…which starts not far from here. The Introduction should state the problem and the experimental remedy without too much detail. That can be presented in the Methods and Discussion.

(6) Line 187. Why not move your description of the SMSCG here?

(7) Line 234. Table 1 has considerable information. The question is whether to put it in the manuscript or as an addendum. I would learn toward recommending that it is placed within the text of ms because the information is so important. Another reviewer might disagree with me.

(8) Line 463. “Field sampling supported our overall prediction that the Flow Action would allow Delta Smelt to colonize Suisun Marsh”. How many Delta Smelt were caught in Suisun Marsh? Doesn’t seem like many, indicated in Figure 10. Is this really a significant number to justify the above statement. Perhaps, the statement should be a more qualified one, based upon the result that some were caught, and none were caught before or after the flow manipulation. How about saying something like, “Although few smelt were caught throughout the Delta, some were caught in Suisun Marsh during the Flow Action. This is consistent with the action having an effect on the recruitment of Delta Smelt”.

Seems to me a huge amount of good work went into showing a result that is somewhat questionable. When contacted by the press about whether white shark attacks are increasing based on two or three more during a particular year when there may have been none on the prior year, I have always presented that caveat that the sample size is so small that it may be a random process instead of a true increase. You are presented with the same dilemma working with numbers that are so small that one wonders whether an increase is a true one.

What about prior years? Have Delta smelt been caught in Suisun Marsh or near Suisun Marsh at other times? If so, I would be good to have a plot in the Discussion of the numbers sampled in years before and after. This would place the results in the context of historical observations.

(9) Line 474. The authors dwell on the results of the study right at the beginning of the Discussion. Why not place the historical information here that was at the beginning of the Introduction. Then talk about the results of this study in comparison to those of other studies. It seems that some of the Discussion is redundant with the Results. Can this section be shortened by not repeating what was included in the Results?

General Comments

(1) The Introduction is very comprehensive and well related to the existing scientific literature for the region. Yet it is very long, so can some of the information be moved to the Discussion. This is really a decision for the authors. Do readers need all of this information prior to learning about the Methods and Results of the study or can some of it be stated in the Discussion? Generally, the Introductions to papers are no more than four to five paragraphs at most.

(2) This study was very comprehensive, and also experimental…and for this, the participants should be complimented. Whether it indicates that the modification is truly effective in permitting more Delta smelt to colonize the Suisun Marsh seems questionable. …although there is some support! If the experiment were conducted another time…and that would require an extraordinary effort to double the N…would they also capture Delta Smelt in Suisun Marsh during the Flow Action and not at other times.

(3) Again, another reason for publishing this in PLoS ONE despite it being good science, is the multidisciplinary nature of the study should be show-cased for others to follow in future studies. It should encourage others to make experimental manipulations of the environment and monitor their effect on biological systems.

Conclusion

(1) The main value to publishing this study in PLoS ONE is heuristic one, that is to encourage this approach to studying biological systems. Although I’m not totally convinced that they have demonstrated the Flow Action had an effect on Delta Smelt recruitment to the Suisun Marsh…there is some evidence to that effect.

Recommendation

(1) Major Revisions,

…although they really only consist of (1) improving a map, (2) moving some text around, (3) being cautious about concluding that the Flow Action created significant recruitment of Delta Smelt in the Delta, and 4) cutting some redundant text from the Discussion. For a paper of this complexity, the revisions are few.

I only review openly, and the authors are welcome to contact me to discuss my remarks.

<*{{{{{< <*{{{{{< <*{{{{{<

<*{{{{{< <*{{{{{<

A. Peter (Pete) Klimley, Ph.D. Adjunct Professor, Retired

Biotelemetry Consultants Director, Retired, Biotelemetry Laboratory

2870 Eastman Lane Dept. of Wildlife, Fish, & Conserv. Biology

Petaluma, California 94952 Univ. of California, Davis, California 95616

Phone: (707) 481-1547 (Cellular)

Dr. Hammerhead: www.pbs.org/wgbh/nova/sharks/masters/hammerhead.html

Associate Editor, Animal Biotelemetry: http://www.animalbiotelemetry.com/logon?url=%2Fmy

Associate Editor: PLOS ONE: http://www.editorialmanager.com/pone/default.asp

Associate Editor: Environmental Biology of Fishes: https://www.springer.com/journal/10641

Author: “Dr. Hammerhead Swims with Sharks”, Fins Attached, 260 pp, later 2019

" Biology of Sharks and Rays", Univ. of Chicago Press, Chicago, 512 pp, 2013

"Secret Life of Sharks", Simon and Schuster, New York, 292 pp, 2003

Editor: "Great White Sharks: The Biology of Carcharodon carcharias",

Academic Press, 517 pp, 1996

<*{{{{{< <*{{{{{< <*{{{{{<

<*{{{{{< <*{{{{{<

Reviewer #2: This paper explores physical and biological effects of directing pulses of low salinity water into a tidal marsh. The question is worthy of study because tidal estuaries are important habitat for many sensitive fishes, and many estuaries are at risk from sea level rise and flow modifications upstream. The unique radial gates in a channel leading to the estuary marsh provided the authors an opportunity to explore the potential benefits of a flow action. Although the results are specific to that particular estuary, they may be of interest to coastal resource managers in other settings.

The manuscript is technically sound regarding the experimental design and, for the most part, the conclusions drawn. The statistical analysis is not obviously inappropriate, but could be made more rigorous to support stronger conclusions, and in some cases requires additional explanation and check of assumptions. The authors have stated that the data or links to the data are available in the text, but I did not find them. The URLs for online databases are provided, but these are larger databases from which the subset of data used are drawn, not the data sets used specifically for this manuscript. Data collected for this dataset in particular are not obviously available.

The paper is generally well-written. I would like to see some more consideration of the analytical methods and a bit more detail in the methods description, and a small restructuring of the results. Additionally, it would be helpful to expand the discussion of the potential impacts of this sort of flow action for recovery of the Delta Smelt population. Given the observed extent of salinity differences and the observed Delta Smelt catch, what is the potential effect of this flow action on the population as a whole? Should this type of flow action and salinity control be implemented elsewhere in the Delta? These issues are especially appropriate considering the manuscript’s title.

The linear model analysis of the 2018 continuous water quality parameters (salinity, temperature, chlorophyll, and turbidity) is reasonable for showing temporal and spatial differences in these parameters, and it is reasonable to hypothesize that differences for the marsh locations in August are due to the flow action. The river station acts as a necessary spatial control. However, without comparison to a temporal control as well (i.e., historical data), it is not possible to actually conclude that differences are due to the flow action rather than to seasonal changes. Eyeballing the monthly averages and assuming a monotonic increase or decrease through the season is insufficient. Neither is it possible to conclude that any lack of differences reflects a limitation of the flow action; for example, might it be conceivable that temperature would have shown a spatial difference without the flow action, but by introducing river water to the marsh, the temperature difference was lost? You have daily historical data for temperature and salinity, so it should be possible to do some sort of BACI analysis using the historical data as temporal controls. As it is, the historical data are used only in a descriptive graphical analysis that seems more appropriate for a first step but not for a final analytical step. Can you expand your linear model in line 287 to include historical data with a year effect and a treatment effect? If you wish to use only historical data from years that are not markedly different from 2018, you can select those years by identifying the cluster where 2018 falls in the hierarchical cluster analysis of July salinity. That may simply be the “dry” cluster. The lack of historical data for chlorophyll, turbidity, and zooplankton (and Delta smelt) limit your ability to conclude a treatment effect, or lack thereof, for those metrics.

In the results, it would aid reader comprehension to make clearer that you are presenting (1) general comparisons between Suisun Marsh and the River, in the absence of the flow action, (2) predicted effects of the flow action from a hydrodynamic model, and (3) observed effects of the flow action.

Lines 34-36: It is surprising to see that “small numbers of Delta Smelt” support a hypothesis of “benefit” to that species. As I read your paper, I understood that the “small numbers” were in the context of even lower numbers in the surrounding months, but that was not obvious in the Abstract. Perhaps clarify.

Lines 69-70 and Figure 1: Are the dashed red lines indicating the gates? Or differentiating between regions? It is difficult to see that the SMSCG is actually in a water channel (i.e., Montezuma Slough). Can you make the slough clearer on the map, and also Grizzly Bay? It might be helpful to zoom in a bit more, if possible, and include the larger setting as an inset. I think you also need an inset that places this map in the context of the state of California and/or USA. The figure could use a legend. Where is the Low Salinity Zone, and the zooplankton/Microcystis sampling sites? For those sampling sites, it might be sufficient to add latitude and longitude to the map.

Lines 86-87: What factors are thought to make Suisun Marsh and Suisun Bay suitable rearing areas? E.g., low salinity, high turbidity, etc.

Lines 112-113: Does this mean that Suisun Marsh would normally already be turbid during your study? So that lowering salinity levels during this time would then make the habitat suitable for Delta Smelt? This is noted in the discussion, but it would be helpful to say that here, too.

Lines 122-126: How does the SMSCG result in flow augmentation into the Marsh? I understand how it would lower salinity, but not how it increases inflow.

Line 128: It would be helpful to clearly state what “upstream” and “downstream” mean in relation to Suisun Marsh, Montezuma Slough, Grizzly Bay, and Suisun Bay. For example, for Suisun Bay, is “upstream” used to denote east into the Sacramento and San Joaquin rivers, or north into Grizzly Bay and Suisun Marsh?

Lines 138-140 and Figure 2: What does “downstream” mean for Montezuma Slough? Is it reasonable to identify the flashboards, boat lock, and jetty referred to in lines 332-336 here?

Lines 154-160: Is Microcystis part of your hypotheses? Also, some of these hypotheses seem related to the flow action while others are not. Perhaps reorder to place those referring to general patterns first, and those referring to effects of the flow action second. It would be helpful if you referred to these hypotheses in your results and/or discussion.

Lines 156-157: how would you know if there is a treatment effect, given that you have no historical data for turbidity or chlorophyll?

Lines 158-159: Is this pattern meant to be related to the flow action?

Lines 160-161: How is this hypothesis to be tested? To conclude “allows” requires demonstrating the Delta Smelt were unable to use the region in the absence of the flow action, and were able to use the action in the presence of the flow action. Can this hypothesis be rephrased in testable or quantifiable format: “…would result in increased numbers of Delta Smelt” or “… would result in salinity conditions through to be suitable for rearing Delta Smelt (i.e., <6 psu).

Line 173: What input did the hydrodynamic model require? For example, did it use real-time observations of salinity, velocity, and other conditions in Suisun Marsh and Grizzly Bay, or did it require only measures such as Delta inflow and outflow data?

Line 186: Are there seasonal changes in the geographic extent of the LSZ? Or is the LSZ well-defined geographically? If so, add it to Figure 1.

Line 201: Do you mean 2003-2017 (i.e., did you include 2018 in the hierarchical cluster analysis)?

Lines 200-207: This paragraph is in the Data Sources section but (1) describes how data are used, and (2) does not mention the source or nature of the salinity data. The paragraph should be moved to the data analysis section.

Line 210: What protocol did you use to convert water velocity data to tidal flow?

Line 211: “estimated flows in Montezuma Slough” – how did you estimate these? By observation or by using the hydrodynamic model output?

Lines 214-215: Clarify that specific conductivity is used to represent salinity.

Line 227: the link is broken.

Lines 272 (Data Analysis): This first sentence (lines 274-275) is very helpful for orienting the reader. It would be helpful to give one more sentence on how you used the hydrodynamic simulation modeling – was it used separately from the observation data, or did it depend on those data? In other words, it appears to me that your results are based on two overall approaches: simulation modeling to predict what you think would or did happen with the flow action, and observation data examining what did happen. The observation data are necessarily on a much cruder spatial scale than the simulation modeling, but they provide empirical evidence of conditions, whereas the simulation modeling is only hypothesized conditions.

Lines 275-276: Your hypotheses are based on comparisons with historical data; analysis that is more than visual comparison of monthly averages is required to investigate those hypotheses. For salinity and temperature, you have daily or 15-minute event data, so you could use a linear model to look for a seasonal effect that is different in the flow action than in historical years. Or, if you continue to use the monthly average, test whether the observed 2018 monthly average is different than expected in the absence of the flow action.

Lines 281-283: Please clarify that these linear models use only data from within 2018, and note that they cannot test for a definitive flow action effect because they do not compare to historical data.

Lines 287, 304: What is the error structure? Additive normal errors? Did you examine residuals and modeling assumptions? For example, it appears that you are assuming the same magnitude of error exists among sites in both regions – did you test or examine residuals for that? Did you consider spatial autocorrelation, or are the sites far enough apart to make it unnecessary (i.e., are the data points actually independent)?

Lines 304-305: This approach to modeling ordinal data such as Microcystis requires that the difference between the qualitative levels be uniform across the scale. For example, the difference between levels 1 and 2 must be the same as the difference between levels 4 and 5. Is that, in fact, reasonable here? It is difficult for the reader to assess. Alternative analysis would be an ordered logit or ordered probit model.

Lines 314-317: Please provide some additional details on the methods here, as well as some references.

Line 322+: Please provide more context for the habitat modeling. What type of input does the UnTRIM model use? Is it only Delta inflow and/or outflow? Flow from other stations? Velocity? Presumably it does not use salinity observations as input, but provides salinity measures as output.

Lines 339-340: How did you evaluate it? Did you use the same methods as previously described, and if so, which of them? More detail is needed here.

Line 346: Should “were” be changed to “was”?

Lines 348-351: Does the flow augmentation account for the higher than expected September NDOI in 2018 compared to the dry years?

Lines 353-354 and Figure 3: Why not use the monthly average daily NDOI? That would have given you standard errors on the monthly mean, not only standard deviation, and would have provided a more useful comparison of 2018 with other years.

Lines 369-373 and Figure 4: Similar to comment for Figure 3: By comparing monthly averages using standard deviation, it is reasonable to compare 2018 (1 data point) to previous years (3 data points each, yielding monthly mean and SD) – although less reasonable seeing that you actually had daily data. But it is not reasonable to compare 2018 data points from different months, because you have no measure of dispersion for the 2018 bars with only one monthly data point per bar. You could compare the monthly mean of daily averages with SE of mean, if you want to compare between months of 2018 (see lines 360-361).

Line 375: It would be helpful to clarify that this is only 2018 data in this table. Also, please indicate the degrees of freedom for the t-tests.

Table 1: What is the difference between a p-value of 0.000 and a p-value of <0.0001?

Table 1, Zooplankton: Where are the Region x Month terms?

Lines 383-398: Do the modeling results reflect reality (i.e., are they dependent at all on actual salinity observations in the marsh, aside from initial calibration), or could you have gotten this model result before doing any actual flow augmentation? If the latter, that is fine, but it would be helpful to clarify so readers can judge how much dependence to place on the modeling results. Also, if the modeling results were used to predict effects of the flow action (rather than to observe those effects), then it makes sense to present the modeling results before the data results.

Line 429: Whether the shift was due to the flow action vs seasonal progression is not obvious.

Lines 429-430 (PERMANOVA) and Table 2: We see a difference with month and with region, but not that it was different during all three months and in all three regions – it could be just one month and/or region that makes the difference. Also, did you perform the PERMANOVA on chlorophyll and turbidity?

Figure 7: Use the same vocabulary in figure 7 as in the text (“chlorophyll”). Can you use colors that are more easily distinguishable when printed in grayscale?

Lines 444-445: October looks different in Figure 8. Do you mean only July – September?

Figure 9: Do the differences between the levels mean the same thing across the entire scale? Otherwise, it is hard to interpret a SD on an ordinal measure.

Lines 478-479: This paragraph appears to address estuaries in general; are there references available from systems other than the SFE?

Lines 520-523: So, by lowering salinity, the flow action allows Delta Smelt to move into habitat that is otherwise suitable for them based on turbidity.

Lines 525-536: The lack of presumed effect on temperature may be explored more usefully by comparing to historical data.

Lines 585-588: Noted, but it appears that a flow action tailored to the site and system has potential to affect habitat over an extended period.

6. PLOS authors have the option to publish the peer review history of their article (what does this mean?). If published, this will include your full peer review and any attached files.

Reviewer #1: **Yes: **Abbott Peter Klimley

Reviewer #2: No

---

## [Author Response · Author response to Decision Letter 0]

5 Aug 2020

As noted above, we have uploaded a Responses to Comments file. The reponses to specific reviewer and editor comments are also pasted below. 

Reviewers' comments:

Comments to the Author

1. Is the manuscript technically sound, and do the data support the conclusions?

Reviewer #1: Partly

Reviewer #2: Yes

Response: To address Reviewer 1’s concern, we have tempered our conclusions about the population effects on Delta Smelt, and added an additional analysis using historical data.

2. Has the statistical analysis been performed appropriately and rigorously?

Reviewer #1: Yes

Reviewer #2: Yes

Response: The statistical analyses have been updated in response to Reviewer 2’s detailed comments (below).

3. Have the authors made all data underlying the findings in their manuscript fully available?

Reviewer #1: Yes

Reviewer #2: No

Response: To address Reviewer 2’s comment, we have published the complete data set on the Environmental Data Initiative (EDI) open data website. Here is the citation: 

Hartman, R.K., T. Sommer, M. Koohafkan, C. Burdi, A. Bever, M. MacWilliams, and J. Galef. 2020. Interagency Ecological Program: Water quality, fish, and zooplankton monitoring and modeling to support the 2018 Suisun Marsh Salinity Control Gates Summer Action ver 2. Environmental Data Initiative. https://doi.org/10.6073/pasta/72d4abd5c679260d0130655d1179e47b (Accessed 2020-08-04).

4. Is the manuscript presented in an intelligible fashion and written in standard English?

Reviewer #1: Yes

Reviewer #2: Yes

Response: We have addressed the editorial suggestions provided by the reviewers.

Reviewer #1: Specific Comments

(1) Line 69. Can more detail be presented in Figure 1 such as the water course, on which the Salinity Control Gates are situated? The labels appear fuzzy to me, but perhaps that is due to the figure being converted to pdf format. This map is really important and should have sufficient detail for understanding the study. Why not start your map quadrat at Carquinez Strait so that you can provide greater detail? If you want, put an insert in the corner with San Francisco Bay. Yet, is it really important to show San Francisco Bay?

Response: The map has been updated to provide a bit more detail. We chose to include San Francisco Bay because it is a core part of the SFE, and will help outside readers better understand the location of the study area.

(2) Line 72. Why not start the Introduction here and place the proceeding information at the beginning of the Introduction?

Response: We agree that it is helpful to have a concise Introduction and have worked to reduce its length (see below). However, these opening paragraphs are critical to set the overall context for the study. Our opinion is that the first sections of the text should provide the broader ecological background, something that would be lost if we started by discussing Delta Smelt, a species of local concern.

(3) Line 114. Could this paragraph be moved into the Methods section? Again, I am trying to shorten the introduction so that the reader does not lose interest due to its length.

Response: As requested, we moved this section to the Methods 

(4) Line 138. This is a very illustrative figure. It gives the reader a good idea of how salinity is regulated. Again, it would be better placed in the Methods.

Response: As requested, we moved the Figure to the Methods. 

(5) Line 142. Start up again with this paragraph. You would have to add some text to briefly describe the SMSCG here, but describe it in detail in the Methods…which starts not far from here. The Introduction should state the problem and the experimental remedy without too much detail. That can be presented in the Methods and Discussion.

Response: As noted above, incorporated this suggestion. 

(6) Line 187. Why not move your description of the SMSCG here?

Response: Done. 

(7) Line 234. Table 1 has considerable information. The question is whether to put it in the manuscript or as an addendum. I would learn toward recommending that it is placed within the text of ms because the information is so important. Another reviewer might disagree with me.

Response: We agree that this information is probably best suited for the Methods. 

(8) Line 463. “Field sampling supported our overall prediction that the Flow Action would allow Delta Smelt to colonize Suisun Marsh”. How many Delta Smelt were caught in Suisun Marsh? Doesn’t seem like many, indicated in Figure 10. Is this really a significant number to justify the above statement. Perhaps, the statement should be a more qualified one, based upon the result that some were caught, and none were caught before or after the flow manipulation. How about saying something like, “Although few smelt were caught throughout the Delta, some were caught in Suisun Marsh during the Flow Action. This is consistent with the action having an effect on the recruitment of Delta Smelt”.

Response: This is a good point. We adjusted our results to acknowledge that we cannot describe the population effects of the Flow Action with any confidence, and that our observations are primarily qualitative. We also added a new statistical analysis using historical Smelt catch data that helps support our conclusions. 

Seems to me a huge amount of good work went into showing a result that is somewhat questionable. When contacted by the press about whether white shark attacks are increasing based on two or three more during a particular year when there may have been none on the prior year, I have always presented that caveat that the sample size is so small that it may be a random process instead of a true increase. You are presented with the same dilemma working with numbers that are so small that one wonders whether an increase is a true one.

What about prior years? Have Delta smelt been caught in Suisun Marsh or near Suisun Marsh at other times? If so, I would be good to have a plot in the Discussion of the numbers sampled in years before and after. This would place the results in the context of historical observations.

Response: We added an analysis using data from prior years to help put the 2018 results into a historical perspective. 

(9) Line 474. The authors dwell on the results of the study right at the beginning of the Discussion. Why not place the historical information here that was at the beginning of the Introduction. Then talk about the results of this study in comparison to those of other studies. It seems that some of the Discussion is redundant with the Results. Can this section be shortened by not repeating what was included in the Results?

Response: We are not sure about how to address this request. The opening paragraphs of the Discussion don’t actually repeat the results. Instead, they provide a broader ecological context for our investigation. We do describe some of the results in later paragraphs, but we believe that one of the roles of the Discussion is to explain to Results. We can’t explain the Results without some mention of the general patterns. 

General Comments

(1) The Introduction is very comprehensive and well related to the existing scientific literature for the region. Yet it is very long, so can some of the information be moved to the Discussion. This is really a decision for the authors. Do readers need all of this information prior to learning about the Methods and Results of the study or can some of it be stated in the Discussion? Generally, the Introductions to papers are no more than four to five paragraphs at most.

Response: We followed the reviewer’s guidance and shortened the Introduction. 

(2) This study was very comprehensive, and also experimental…and for this, the participants should be complimented. Whether it indicates that the modification is truly effective in permitting more Delta smelt to colonize the Suisun Marsh seems questionable. …although there is some support! If the experiment were conducted another time…and that would require an extraordinary effort to double the N…would they also capture Delta Smelt in Suisun Marsh during the Flow Action and not at other times.

Response: We tempered our conclusions about Delta Smelt based on this input.

(3) Again, another reason for publishing this in PLoS ONE despite it being good science, is the multidisciplinary nature of the study should be show-cased for others to follow in future studies. It should encourage others to make experimental manipulations of the environment and monitor their effect on biological systems.

Response: No response required. 

Conclusion

(1) The main value to publishing this study in PLoS ONE is heuristic one, that is to encourage this approach to studying biological systems. Although I’m not totally convinced that they have demonstrated the Flow Action had an effect on Delta Smelt recruitment to the Suisun Marsh…there is some evidence to that effect.

Response: Again, we tempered our conclusions for Delta Smelt, and added a new analysis based on historical data. 

Recommendation

(1) Major Revisions,

…although they really only consist of (1) improving a map, (2) moving some text around, (3) being cautious about concluding that the Flow Action created significant recruitment of Delta Smelt in the Delta, and 4) cutting some redundant text from the Discussion. For a paper of this complexity, the revisions are few.

Response: Thank you for the feedback. Our revisions cover each of these three major requests.

Reviewer #2:

This paper explores physical and biological effects of directing pulses of low salinity water into a tidal marsh. The question is worthy of study because tidal estuaries are important habitat for many sensitive fishes, and many estuaries are at risk from sea level rise and flow modifications upstream. The unique radial gates in a channel leading to the estuary marsh provided the authors an opportunity to explore the potential benefits of a flow action. Although the results are specific to that particular estuary, they may be of interest to coastal resource managers in other settings.

The manuscript is technically sound regarding the experimental design and, for the most part, the conclusions drawn. The statistical analysis is not obviously inappropriate, but could be made more rigorous to support stronger conclusions, and in some cases requires additional explanation and check of assumptions. The authors have stated that the data or links to the data are available in the text, but I did not find them. The URLs for online databases are provided, but these are larger databases from which the subset of data used are drawn, not the data sets used specifically for this manuscript. Data collected for this dataset in particular are not obviously available.

Response: All the data that are presented in the manuscript have been collated into a single data set, which has been published on the Environmental Data Initiative data repository:

https://portal.edirepository.org/nis/mapbrowse?scope=edi&identifier=581&revision=1

The paper is generally well-written. I would like to see some more consideration of the analytical methods and a bit more detail in the methods description, and a small restructuring of the results. Additionally, it would be helpful to expand the discussion of the potential impacts of this sort of flow action for recovery of the Delta Smelt population. Given the observed extent of salinity differences and the observed Delta Smelt catch, what is the potential effect of this flow action on the population as a whole? Should this type of flow action and salinity control be implemented elsewhere in the Delta? These issues are especially appropriate considering the manuscript’s title.

Response: We tried to follow the detailed guidance (below) to address the need for more information about methods, and restructuring the results. In addition, we added text to the Discussion about the population effects of this action.

The linear model analysis of the 2018 continuous water quality parameters (salinity, temperature, chlorophyll, and turbidity) is reasonable for showing temporal and spatial differences in these parameters, and it is reasonable to hypothesize that differences for the marsh locations in August are due to the flow action. The river station acts as a necessary spatial control. However, without comparison to a temporal control as well (i.e., historical data), it is not possible to actually conclude that differences are due to the flow action rather than to seasonal changes. Eyeballing the monthly averages and assuming a monotonic increase or decrease through the season is insufficient. Neither is it possible to conclude that any lack of differences reflects a limitation of the flow action; for example, might it be conceivable that temperature would have shown a spatial difference without the flow action, but by introducing river water to the marsh, the temperature difference was lost? You have daily historical data for temperature and salinity, so it should be possible to do some sort of BACI analysis using the historical data as temporal controls. As it is, the historical data are used only in a descriptive graphical analysis that seems more appropriate for a first step but not for a final analytical step. Can you expand your linear model in line 287 to include historical data with a year effect and a treatment effect? If you wish to use only historical data from years that are not markedly different from 2018, you can select those years by identifying the cluster where 2018 falls in the hierarchical cluster analysis of July salinity. That may simply be the “dry” cluster. The lack of historical data for chlorophyll, turbidity, and zooplankton (and Delta smelt) limit your ability to conclude a treatment effect, or lack thereof, for those metrics.

Response: We have included new models of salinity and temperature in the East Marsh, where the effect of the gate action was greatest. These analyses show that salinity increased less in August than in previous dry years or previous wet years, in line with our graphical results.

In the results, it would aid reader comprehension to make clearer that you are presenting (1) general comparisons between Suisun Marsh and the River, in the absence of the flow action, (2) predicted effects of the flow action from a hydrodynamic model, and (3) observed effects of the flow action.

Response: As requested, we added a short introductory paragraph in the Results to highlight the different categories of that that were evaluated. We also added additional details clarifying that the hydrodynamic model was used to simulation and compare conditions with and without the Flow Action.

Lines 34-36: It is surprising to see that “small numbers of Delta Smelt” support a hypothesis of “benefit” to that species. As I read your paper, I understood that the “small numbers” were in the context of even lower numbers in the surrounding months, but that was not obvious in the Abstract. Perhaps clarify.

Response: We updated the text to temper our conclusions, and added a new analysis of the response of Delta Smelt based on historical observations (Fig 11).

Lines 69-70 and Figure 1: Are the dashed red lines indicating the gates? Or differentiating between regions? It is difficult to see that the SMSCG is actually in a water channel (i.e., Montezuma Slough). Can you make the slough clearer on the map, and also Grizzly Bay? It might be helpful to zoom in a bit more, if possible, and include the larger setting as an inset. I think you also need an inset that places this map in the context of the state of California and/or USA. The figure could use a legend. Where is the Low Salinity Zone, and the zooplankton/Microcystis sampling sites? For those sampling sites, it might be sufficient to add latitude and longitude to the map.

Response: We have developed a new map showing greater detail in Suisun Marsh with regions circled more clearly, included an inset of the study location within the state of California, and included a supplemental table with GPS coordinates for sampling sites. The Low Salinity Zone changed over the course of the Action, so cannot be shown on a single map, instead we have plotted the distribution of the salinity field in figure 5.

Lines 86-87: What factors are thought to make Suisun Marsh and Suisun Bay suitable rearing areas? E.g., low salinity, high turbidity, etc.

Response: We added a sentence describing some of the most important habitat attributes in the Suisun Region.

Lines 112-113: Does this mean that Suisun Marsh would normally already be turbid during your study? So that lowering salinity levels during this time would then make the habitat suitable for Delta Smelt? This is noted in the discussion, but it would be helpful to say that here, too.

Response: This point is already addressed in the Hypotheses, which follows shortly thereafter in the text. We could add it here too, but it seems a bit redundant and counter to Reviewer 1’s recommendation to shorten the Introduction.

Lines 122-126: How does the SMSCG result in flow augmentation into the Marsh? I understand how it would lower salinity, but not how it increases inflow.

Response: We agree, and believe the confusion is caused by our usage of the word “directing”. Water naturally flows from the river into Montezuma Slough during ebb tide and from the bay into Montezuma Slough during flood tide (so the net directional flow over a complete tidal cycle is essentially zero). The gates act to prevent seawater flowing up into Montezuma slough during flood tide, resulting in a net directional flow from the river into Montezuma Slough. We changed the word “directing” to “allowing” and the sentence now reads: “The SMSCG control salinity by allowing low salinity water from the Sacramento River into Montezuma Slough during ebb (outgoing) tides but restricting the tidal flow of higher salinity water into Montezuma Slough during flood (incoming) tides. This strategy generates a net flow of low salinity water from east to west in Montezuma Slough…” Hence, this increase in net flow results in flow augmentation into the Marsh.

Line 128: It would be helpful to clearly state what “upstream” and “downstream” mean in relation to Suisun Marsh, Montezuma Slough, Grizzly Bay, and Suisun Bay. For example, for Suisun Bay, is “upstream” used to denote east into the Sacramento and San Joaquin rivers, or north into Grizzly Bay and Suisun Marsh?

Response: We added additional text to help clarify the locations of “upstream” and “downstream”.

Lines 138-140 and Figure 2: What does “downstream” mean for Montezuma Slough? Is it reasonable to identify the flashboards, boat lock, and jetty referred to in lines 332-336 here?

Response: See previous response for the definition of “downstream”. We updated Figure 2 as requested.

Lines 154-160: Is Microcystis part of your hypotheses? Also, some of these hypotheses seem related to the flow action while others are not. Perhaps reorder to place those referring to general patterns first, and those referring to effects of the flow action second. It would be helpful if you referred to these hypotheses in your results and/or discussion.

Response: As requested, we added Microcystis to the Hypotheses. For the second part of this comment, we consider all of the hypotheses to be related to the Flow Action, not just a subset of them. Hence, we prefer to keep the order of the hypotheses as originally provided.

Lines 156-157: how would you know if there is a treatment effect, given that you have no historical data for turbidity or chlorophyll?

Response: We did not predict there would be a change in chlorophyll or turbidity due to the action. Our goal in measuring chlorophyll and turbidity was to show that Suisun had higher productivity and better habitat characteristics than the River. The Action allowed fish to colonize Suisun by changing salinity, not turbidity or chlorophyll.

Lines 158-159: Is this pattern meant to be related to the flow action?

Response: See previous response for 154-160.

Lines 160-161: How is this hypothesis to be tested? To conclude “allows” requires demonstrating the Delta Smelt were unable to use the region in the absence of the flow action, and were able to use the action in the presence of the flow action. Can this hypothesis be rephrased in testable or quantifiable format: “…would result in increased numbers of Delta Smelt” or “… would result in salinity conditions through to be suitable for rearing Delta Smelt (i.e., <6 psu).

Response: Done

Line 173: What input did the hydrodynamic model require? For example, did it use real-time observations of salinity, velocity, and other conditions in Suisun Marsh and Grizzly Bay, or did it require only measures such as Delta inflow and outflow data?

Response: We added additional details regarding the model domain extent for the hydrodynamic model to make it more clear that we are using a 3D numerical model of the entire San Francisco Bay and Sacramento-San Joaquin Delta. Additional detail on the hydrodynamic model is provided in the Habitat Modeling section. We also directly state what changes to the hydrodynamic model were made between the Flow Action and without Flow Action model scenarios and clarifying that the primary model boundary conditions are significantly distant from the study area and unaffected by the Flow Action. Real-time observations in the study area were used for hydrodynamic model validation but not as model boundary conditions.

Line 186: Are there seasonal changes in the geographic extent of the LSZ? Or is the LSZ well-defined geographically? If so, add it to Figure 1.

Response: The magnitude and location of the LSZ varies substantially depending on season and hydrology, so we did not add it to Figure 1. However, we added text to note the general region that it covers. 

Line 201: Do you mean 2003-2017 (i.e., did you include 2018 in the hierarchical cluster analysis)?

Response: This was a typo. We amended the sentence to read “We first did a hierarchical cluster analysis for 2003 – 2017…”.

Lines 200-207: This paragraph is in the Data Sources section but (1) describes how data are used, and (2) does not mention the source or nature of the salinity data. The paragraph should be moved to the data analysis section.

Response: We think that this paragraph is most appropriate here since it explains how we chose the historical years that were used. 

Line 210: What protocol did you use to convert water velocity data to tidal flow?

Response: : We agree, this was not adequately described. We revised this sentence as follows:

“Water flow through Suisun Marsh (through Montezuma Slough) was computed from Acoustic Doppler Current Profiler (ADCP) measurements of water velocity at National Steel (Figure 1) using the index velocity method [36].”

Line 211: “estimated flows in Montezuma Slough” – how did you estimate these? By observation or by using the hydrodynamic model output?

Response: We agree that this was not totally clear. Flows were calculated based on ADCP measurements, not estimated from model outputs. We removed the second usage of the word “estimated” to clarify that we are referring to the measured flow data described in the preceding sentence. We also clarified the ADCP methodology in response to the previous comment regarding Line 210.

Lines 214-215: Clarify that specific conductivity is used to represent salinity.

Response: Done

Line 227: the link is broken.

Response: DWR’s website changed, new link: https://emp.baydeltalive.com/projects/11285

Lines 272 (Data Analysis): This first sentence (lines 274-275) is very helpful for orienting the reader. It would be helpful to give one more sentence on how you used the hydrodynamic simulation modeling – was it used separately from the observation data, or did it depend on those data? In other words, it appears to me that your results are based on two overall approaches: simulation modeling to predict what you think would or did happen with the flow action, and observation data examining what did happen. The observation data are necessarily on a much cruder spatial scale than the simulation modeling, but they provide empirical evidence of conditions, whereas the simulation modeling is only hypothesized conditions.

Response: We added additional details about the hydrodynamic model and how it was used in this study. We clarified that the hydrodynamic model was used to simulate conditions both with and without the Flow Action, which allowed for direct estimates of the effects of the Flow Action on conditions in Montezuma Slough. We also clarified that the observed data was used with linear models to evaluate conditions before, during, and after the action periods.

Lines 275-276: Your hypotheses are based on comparisons with historical data; analysis that is more than visual comparison of monthly averages is required to investigate those hypotheses. For salinity and temperature, you have daily or 15-minute event data, so you could use a linear model to look for a seasonal effect that is different in the flow action than in historical years. Or, if you continue to use the monthly average, test whether the observed 2018 monthly average is different than expected in the absence of the flow action.

Response: We have added additional analyses comparing 2018 to historical years for salinity and temperature, and added a caveat to our description of the models that only use 2018 data.

Lines 281-283: Please clarify that these linear models use only data from within 2018, and note that they cannot test for a definitive flow action effect because they do not compare to historical data.

Response: We have added the sentence “Because we only have the complete data set for a single year, we cannot conclusively say that the flow action, and not seasonal changes, caused the resulting trends. However, these models can be discussed in regards to our expectations for seasonal and gate-action-related changes.

“

Lines 287, 304: What is the error structure? Additive normal errors? Did you examine residuals and modeling assumptions? For example, it appears that you are assuming the same magnitude of error exists among sites in both regions – did you test or examine residuals for that? Did you consider spatial autocorrelation, or are the sites far enough apart to make it unnecessary (i.e., are the data points actually independent)?

Response: Yes, we checked all modeling assumptions, including homogeneity of variance and independence given a normal distribution. We have added language clarifying our tests.

Lines 304-305: This approach to modeling ordinal data such as Microcystis requires that the difference between the qualitative levels be uniform across the scale. For example, the difference between levels 1 and 2 must be the same as the difference between levels 4 and 5. Is that, in fact, reasonable here? It is difficult for the reader to assess. Alternative analysis would be an ordered logit or ordered probit model.

Response: Because there were relatively few “high” Microcystis scores, we decided it would be more appropriate, and easier to understand, if we converted these values to “presence/absence” and use a binomial regression to model the probability of presence.

Lines 314-317: Please provide some additional details on the methods here, as well as some references.

Response: Further explanation of NMDS and references have been provided.

Line 322+: Please provide more context for the habitat modeling. What type of input does the UnTRIM model use? Is it only Delta inflow and/or outflow? Flow from other stations? Velocity? Presumably it does not use salinity observations as input, but provides salinity measures as output.

Response: We added additional description of the UnTRIM Bay Delta Model, model domain and boundary conditions to the Habitat Modeling section.

Lines 339-340: How did you evaluate it? Did you use the same methods as previously described, and if so, which of them? More detail is needed here.

Response: We added additional description of how the hydrodynamic model simulations were evaluated to the Habitat Modeling Section.

Line 346: Should “were” be changed to “was”?

Response: We think “were” is appropriate since SMSCG is plural (“Suisun Marsh Salinity Control Gates”)

Lines 348-351: Does the flow augmentation account for the higher than expected September NDOI in 2018 compared to the dry years?

Response: The flow augmentation could have accounted for a bit of the higher flows, but the SMSCG Flow Action only occurred during the beginning of September, so the higher 2018 flows were more likely due to other operational considerations in the region.

Lines 353-354 and Figure 3: Why not use the monthly average daily NDOI? That would have given you standard errors on the monthly mean, not only standard deviation, and would have provided a more useful comparison of 2018 with other years.

Response: We agree and implemented this, using the daily NDOI data instead and pooling by the summer classification to compute standard error of the mean for each bar. We included the updated figure here for convenience. The updated figure has also been uploaded into our submission.

Lines 369-373 and Figure 4: Similar to comment for Figure 3: By comparing monthly averages using standard deviation, it is reasonable to compare 2018 (1 data point) to previous years (3 data points each, yielding monthly mean and SD) – although less reasonable seeing that you actually had daily data. But it is not reasonable to compare 2018 data points from different months, because you have no measure of dispersion for the 2018 bars with only one monthly data point per bar. You could compare the monthly mean of daily averages with SE of mean, if you want to compare between months of 2018 (see lines 360-361).

Response: Similar to the NDOI graph, we pooled the daily data to compute standard errors for both the 2018 and historical data and provide standard error bars on all the graphs.

Line 375: It would be helpful to clarify that this is only 2018 data in this table. Also, please indicate the degrees of freedom for the t-tests.

Response: We have clarified the fact that this is only 2018 data

Table 1: What is the difference between a p-value of 0.000 and a p-value of <0.0001?

Response: This was a rounding error. It has been corrected

Table 1, Zooplankton: Where are the Region x Month terms?

Response: We compared models with and without the Region x Month interaction terms using AICc and found them to be similar in predictive power (delta AICc <2), so we presented the simpler model. Text has been updated to reflect this.

Lines 383-398: Do the modeling results reflect reality (i.e., are they dependent at all on actual salinity observations in the marsh, aside from initial calibration), or could you have gotten this model result before doing any actual flow augmentation? If the latter, that is fine, but it would be helpful to clarify so readers can judge how much dependence to place on the modeling results. Also, if the modeling results were used to predict effects of the flow action (rather than to observe those effects), then it makes sense to present the modeling results before the data results.

Response: Hydrodynamic model results are not dependent on any observed water quality data in the vicinity of Suisun Marsh. We expanded the description of the hydrodynamic model to describe the inputs and spatial extent. The exact with and without Flow Action model simulations used in this manuscript could not have been conducted before the Flow Action, because the simulations depend on the meteorology, tidal and non-tidal open boundary water levels, freshwater inflows, and water exports/intakes during the time of the Flow Action, which could not be known a-priori, as well as the specifics of the Flow Action itself, such as the exact timing of the SMSCG summer operation. We clarified in the text that we did previously conduct model simulations of hypothetical flow actions for summer 2005 and 2012, and that these were helpful for the initial planning of the 2018 Flow Action.

Line 429: Whether the shift was due to the flow action vs seasonal progression is not obvious.

Response: We agree. Seasonal effects are complex and may override the effects of a single management action like SMSCG operations. This is a major reason why we used simulation modeling to examine the effects with and without the Flow Action.

Lines 429-430 (PERMANOVA) and Table 2: We see a difference with month and with region, but not that it was different during all three months and in all three regions – it could be just one month and/or region that makes the difference. Also, did you perform the PERMANOVA on chlorophyll and turbidity?

Response: We have included a pairwise post-hoc test to examine differences between regions and months, and corrected the mistake in the table caption (PERMANOVA was run on chlorophyll, turbidity, salinity, and temperature).

Figure 7: Use the same vocabulary in figure 7 as in the text (“chlorophyll”). Can you use colors that are more easily distinguishable when printed in grayscale?

Response: We have changed the label to “chlorophyll” and chosen a color-blind friendly palette.

Lines 444-445: October looks different in Figure 8. Do you mean only July – September?

Response: Yes, statistics were only run on Jul-Sep.

Figure 9: Do the differences between the levels mean the same thing across the entire scale? Otherwise, it is hard to interpret a SD on an ordinal measure.

Response: We have updated the statistical analysis to include presence/absence of microcystis only. The graph is for a qualitative assessment only.

Lines 478-479: This paragraph appears to address estuaries in general; are there references available from systems other than the SFE?

Response: We added some additional references for other estuaries.

Lines 520-523: So, by lowering salinity, the flow action allows Delta Smelt to move into habitat that is otherwise suitable for them based on turbidity.

Response: Yes, that is correct

Lines 525-536: The lack of presumed effect on temperature may be explored more usefully by comparing to historical data.

Response: We have included a historical analysis, and still find no difference between regions. There were some differences between historical years and the action year, but because we only have one “action” year, we were reluctant to read much into it.

Lines 585-588: Noted, but it appears that a flow action tailored to the site and system has potential to affect habitat over an extended period.

Response: Correct. This is what we stated in Lines 583-854.

---

## [Decision Letter · Decision Letter 1]

2 Sep 2020

PONE-D-20-16291R1

Evaluation of a large-scale flow manipulation to the upper San Francisco Estuary: Response of habitat conditions for an endangered native fish

PLOS ONE

Dear Dr. Sommer,

Thank you for submitting your manuscript to PLOS ONE. After careful consideration, we feel that it has merit but does not fully meet PLOS ONE’s publication criteria as it currently stands. Therefore, we invite you to submit a revised version of the manuscript that addresses the points raised by the second reviewer during the review process. Please remember that PLOS ONE has no proof editing services, so pay particular attention at this stage to any grammatical or typographical errors. 

We look forward to receiving your revised manuscript.

Kind regards,

Vanesa Magar, Ph.D.

Academic Editor

PLOS ONE

Reviewers' comments:

Reviewer's Responses to Questions

**Comments to the Author**

1. If the authors have adequately addressed your comments raised in a previous round of review and you feel that this manuscript is now acceptable for publication, you may indicate that here to bypass the “Comments to the Author” section, enter your conflict of interest statement in the “Confidential to Editor” section, and submit your "Accept" recommendation.

Reviewer #1: All comments have been addressed

Reviewer #2: (No Response)

2. Is the manuscript technically sound, and do the data support the conclusions?

Reviewer #1: Yes

Reviewer #2: Yes

3. Has the statistical analysis been performed appropriately and rigorously? 

Reviewer #1: Yes

Reviewer #2: Yes

4. Have the authors made all data underlying the findings in their manuscript fully available?

Reviewer #1: Yes

Reviewer #2: Yes

5. Is the manuscript presented in an intelligible fashion and written in standard English?

Reviewer #1: Yes

Reviewer #2: Yes

6. Review Comments to the Author

Reviewer #1: The two most important improvements I recommended were implemented -- (1) adding more information to the map and (2) qualifying their conclusion regarding the effect on Delta smelt. It is good that they added historical data in Figure 11.

Reviewer #2: The revised manuscript is much improved in presentation, methods, and discussion. I appreciate the work the authors did to address my earlier concerns. There are a few additional wording changes to make, but otherwise I recommend publication.

Lines 73-74, Figure 1: There are no ovals that I see – do you mean triangles? Please explain what the red dashed lines are.

Line 73: Change “are” to “is”.

Line 164 and elsewhere: Should this be “are”, consistent with line 160? The verb tense associated with SMSCG is inconsistent throughout the manuscript; sometimes it is singular and sometimes it is plural.

Line165: Perhaps insert "area" before "experiences", to avoid awkwardness in subject-verb agreement.

Line 165: “flow at tidal time scale”: This is a bit awkward. "flow reversal on a tidal time scale", perhaps.

Line 196: “of three-dimensional”: Change to “of a three-dimensional”.

Line 234: “specific conductivity (as an alternative to salinity)”: Yet you refer to salinity in the results. It is unclear if you are actually referring to specific conductivity when you refer to salinity. If so, then it is not really an "alternative" to salinity but rather a surrogate for salinity.

Line 316: recommend omitting "also".

Line 325: Change “was” to “were”.

Lines 323-331: "Normal additive errors were modeled." (presumably)

Line 341: “with additive normal error structures”: Important to say this for the appropriate models, but it is not entirely correct here, because you used generalized linear models with binomial errors for Microcystis levels.

Line 362: “samples”: of zooplankton and Microcystis?

Line 425: “simulations, visualize the”: awkward phrasing

Line 429: Change “was” to “were”.

Line 435: Change “was using” to “was modeled using”.

Line 442: Change “includes” to “include”.

Line 459: Table 1 does not address regional comparisons.

Line 460: Change “were” to “was”.

Line 463: That is more apparent from Table 1, although restricted to the East Marsh. I am not sure how you are concluding this from Table 2. If it is based on the interaction effects, it is missing the reference to the river component of the interaction effect. That is, the interaction effects are interpreted relative to (1) intercept (July river), (2) month main effect (August river), and (3) region main effect (July marsh, for whichever marsh region is relevant), so that the River main effect in July is part of the interpretation of the Aug*East Marsh or Aug*West Marsh effects. Perhaps: significantly reduce salinities in Suisun Marsh during August as compared to the River in July. Although that is not as neat as what you are aiming for.

Line 464: Table 1 does not obviously demonstrate that. It may be deducible from Table 1, but only with considerable interpretation.

Line 466: Omit “Although”.

Line 469: “closer to historical … September”: This is not obvious from Table 1. It is shown in Fig. 4.

Line 491: Throughout this table (Microcystis), you do not need so many significant digits.

Line 524: “modeling”: Do you mean the simulation modeling?

Line 578: “were not detected”: "none were detected" would be better, to avoid implication that you might catch the same 7 individuals.

Line 583: Change colon to period.

Line 594: Change “B)” to “B:” for consistency.

7. PLOS authors have the option to publish the peer review history of their article (what does this mean?). If published, this will include your full peer review and any attached files.

Reviewer #1: **Yes: **A. Peter Klimley, Ph.D.

Reviewer #2: No

---

## [Author Response · Author response to Decision Letter 1]

8 Sep 2020

Comments to the Author

1. If the authors have adequately addressed your comments raised in a previous round of review and you feel that this manuscript is now acceptable for publication, you may indicate that here to bypass the “Comments to the Author” section, enter your conflict of interest statement in the “Confidential to Editor” section, and submit your "Accept" recommendation.

Reviewer #1: All comments have been addressed

Reviewer #2: (No Response)

Response: No response needed.

2. Is the manuscript technically sound, and do the data support the conclusions?

Reviewer #1: Yes

Reviewer #2: Yes

Response: No response needed.

3. Has the statistical analysis been performed appropriately and rigorously?

Reviewer #1: Yes

Reviewer #2: Yes

Response: No response needed.

4. Have the authors made all data underlying the findings in their manuscript fully available?

Reviewer #1: Yes

Reviewer #2: Yes

Response: No response needed.

5. Is the manuscript presented in an intelligible fashion and written in standard English?

Reviewer #1: Yes

Reviewer #2: Yes

Response: No response needed.

6. Review Comments to the Author

Reviewer #1: The two most important improvements I recommended were implemented -- (1) adding more information to the map and (2) qualifying their conclusion regarding the effect on Delta smelt. It is good that they added historical data in Figure 11.

Response: Glad this addressed your concerns.

Reviewer #2: The revised manuscript is much improved in presentation, methods, and discussion. I appreciate the work the authors did to address my earlier concerns. There are a few additional wording changes to make, but otherwise I recommend publication.

Response: See below for our efforts to address the suggested wording changes.

Lines 73-74, Figure 1: There are no ovals that I see – do you mean triangles? Please explain what the red dashed lines are.

Response: We updated Figure 1 to make it consistent with the legend and text.

Line 73: Change “are” to “is”.

Response: Done

Line 164 and elsewhere: Should this be “are”, consistent with line 160? The verb tense associated with SMSCG is inconsistent throughout the manuscript; sometimes it is singular and sometimes it is plural.

Response: We made the recommended change in this location and other parts of the manuscript.

Line165: Perhaps insert "area" before "experiences", to avoid awkwardness in subject-verb agreement.

Response: We made the suggested edit.

Line 165: “flow at tidal time scale”: This is a bit awkward. "flow reversal on a tidal time scale", perhaps.

Response: We adopted the suggested edit.

Line 196: “of three-dimensional”: Change to “of a three-dimensional”.

Response: Done

Line 234: “specific conductivity (as an alternative to salinity)”: Yet you refer to salinity in the results. It is unclear if you are actually referring to specific conductivity when you refer to salinity. If so, then it is not really an "alternative" to salinity but rather a surrogate for salinity.

Response: We adopted the suggested edit.

Line 316: recommend omitting "also".

Response: Done

Line 325: Change “was” to “were”.

Response: Done

Lines 323-331: "Normal additive errors were modeled." (presumably)

Response: Done

Line 341: “with additive normal error structures”: Important to say this for the appropriate models, but it is not entirely correct here, because you used generalized linear models with binomial errors for Microcystis levels.

Response: We changed the line to read: “we used a second series of linear models on the data from 2018. Continuous water quality parameters and zooplankton used additive normal error structures, Microcystis used a binomial error structure.”

Line 362: “samples”: of zooplankton and Microcystis?

Response: We clarified the sentence based on this suggested addition.

Line 425: “simulations, visualize the”: awkward phrasing

Response: Sentence changed to read: “simulations, and to visualize the”.

Line 429: Change “was” to “were”.

Response: Done

Line 435: Change “was using” to “was modeled using”.

Response: Change made.

Line 442: Change “includes” to “include”.

Response: Done

Line 459: Table 1 does not address regional comparisons.

Response: We deleted the reference to Table 1 here.

Line 460: Change “were” to “was”.

Response: Done

Line 463: That is more apparent from Table 1, although restricted to the East Marsh. I am not sure how you are concluding this from Table 2. If it is based on the interaction effects, it is missing the reference to the river component of the interaction effect. That is, the interaction effects are interpreted relative to (1) intercept (July river), (2) month main effect (August river), and (3) region main effect (July marsh, for whichever marsh region is relevant), so that the River main effect in July is part of the interpretation of the Aug*East Marsh or Aug*West Marsh effects. Perhaps: significantly reduce salinities in Suisun Marsh during August as compared to the River in July. Although that is not as neat as what you are aiming for.

Response: Based on this input, we changed Lines 463-464 to read: As expected, the major effect of the Flow Action was to significantly reduce salinities in Suisun Marsh during August as compared to July (Fig 4; Table 1). The net effect was that August salinity conditions in the Marsh were comparable to historical wet years (Fig 4).

Line 464: Table 1 does not obviously demonstrate that. It may be deducible from Table 1, but only with considerable interpretation.

Response: See previous comment.

Line 466: Omit “Although”.

Response: Done

Line 469: “closer to historical … September”: This is not obvious from Table 1. It is shown in Fig. 4.

Response: We changed the reference to Fig 4 instead.

Line 491: Throughout this table (Microcystis), you do not need so many significant digits.

Response: We updated the table as requested.

Line 524: “modeling”: Do you mean the simulation modeling?

Response: We modified the text as indicated.

Line 578: “were not detected”: "none were detected" would be better, to avoid implication that you might catch the same 7 individuals.

Response: Done

Line 583: Change colon to period.

Response: Done

Line 594: Change “B)” to “B:” for consistency.

Response: Done.

---

## [Editor Report · Decision Letter 2]

17 Sep 2020

Evaluation of a large-scale flow manipulation to the upper San Francisco Estuary: Response of habitat conditions for an endangered native fish

PONE-D-20-16291R2

Dear Dr. Sommer,

We’re pleased to inform you that your manuscript has been judged scientifically suitable for publication and will be formally accepted for publication once it meets all outstanding technical requirements.

Kind regards,

Vanesa Magar, Ph.D.

Academic Editor

PLOS ONE
---

## [Editor Report · Acceptance letter]

21 Sep 2020

PONE-D-20-16291R2

Evaluation of a large-scale flow manipulation to the upper San Francisco Estuary:  Response of habitat conditions for an endangered native fish

Dear Dr. Sommer:

I'm pleased to inform you that your manuscript has been deemed suitable for publication in PLOS ONE. Congratulations! Your manuscript is now with our production department.

Kind regards,

on behalf of

Dr. Vanesa Magar 

Academic Editor

PLOS ONE